# Characterization of small fiber pathology in a mouse model of Fabry disease

Lukas Hofmann[1], Dorothea Hose[1], Anne Grießhammer[1], Robert Blum[2], Frank Döring[3], Sulayman Dib-Hajj[4], Stephen Waxman[4], Claudia Sommer[1], Erhard Wischmeyer[3], Nurcan Üçeyler[1]*

[1]Department of Neurology, University of Würzburg, Würzburg, Germany; [2]Institute of Clinical Neurobiology, University of Würzburg, Würzburg, Germany; [3]Molecular Electrophysiology, Institute of Physiology and Center of Mental Health, University of Würzburg, Würzburg, Germany; [4]Center for Neuroscience and Regeneration Research, Yale Medical School and Veterans Affairs Hospital, West Haven, United States

**Abstract** Fabry disease (FD) is a life-threatening X-linked lysosomal storage disorder caused by α-galactosidase A (α-GAL) deficiency. Small fiber pathology and pain are major FD symptoms of unknown pathophysiology. α-GAL deficient mice (GLA KO) age-dependently accumulate globotriaosylceramide (Gb3) in dorsal root ganglion (DRG) neurons paralleled by endoplasmic stress and apoptosis as contributors to skin denervation. Old GLA KO mice show increased TRPV1 protein in DRG neurons and heat hypersensitivity upon i.pl. capsaicin. In turn, GLA KO mice are protected from heat and mechanical hypersensitivity in neuropathic and inflammatory pain models based on reduced neuronal $I_h$ and $Na_v1.7$ currents. We show that in vitro α-GAL silencing increases intracellular Gb3 accumulation paralleled by loss of $Na_v1.7$ currents, which is reversed by incubation with agalsidase-α and lucerastat. We provide first evidence of a direct Gb3 effect on neuronal integrity and ion channel function as potential mechanism underlying pain and small fiber pathology in FD.
DOI: https://doi.org/10.7554/eLife.39300.001

*For correspondence:
uceyler_n@ukw.de

**Competing interests:** The authors declare that no competing interests exist.

## Introduction

Thermal hyposensitivity and pain based on small nerve fiber pathology are cardinal neurological symptoms of the progressive and life-threatening X-linked Fabry disease (FD) (*Zarate and Hopkin, 2008*). While genetically caused reduction or loss of α-galactosidase A (α-GAL) with consecutive lysosomal accumulation of globotriaosylceramide (Gb3) has long been considered the pathophysiological key event in FD (*Romeo and Migeon, 1970*), the mechanisms underlying sensory impairment, pain, and denervation remain unclear.

Men more than women with FD report reduction in cold and warm sensation over time, which is reflected by increased thermal perception thresholds and paralleled by loss of intraepidermal innervation (*Üçeyler et al., 2011*). Starting in early childhood, patients also report mainly episodic, acral, burning pain triggered by heat, fever, or physical activity (*Üçeyler et al., 2014*). The finding of potential Gb3 deposits in dorsal root ganglion (DRG) neurons (*Gadoth and Sandbank, 1983*; *Lakomá et al., 2016*; *Marshall et al., 2010*) and the presence of peripheral neuropathy mostly of the small fiber type (*Üçeyler et al., 2011*) supports the hypothesis of a pathophysiological mechanism involving Gb3 accumulation.

Having investigated the largest and oldest cohort (>24 months) reported thus far for the α-GAL deficient (GLA KO) mouse model of FD, we previously described an age-dependent development of thermal hyposensitivity mirroring the clinical phenotype (*Üçeyler et al., 2016*). Whether Gb3

**eLife digest** Fabry disease is a life-threatening disorder that runs in families and affects many parts of the body. Symptoms begin in early childhood, often with episodes of burning pain in the hands and feet. As patients with Fabry disease grow older, sensory nerve fibers in their skin start to break down. As a result, affected individuals may often struggle to detect heat or cold against their skin.

Mutations in a gene called alpha-galactosidase A cause Fabry disease. These mutations prevent the alpha-galactosidase A (alpha-GAL) enzyme from working properly. This enzyme breaks down fatty substances in the cells, in particular a molecule named globotriaosylceramide (Gb3). In patients with Fabry disease, Gb3 accumulates inside cells and is thought to cause pain, reduced temperature sensitivity, and loss of nerve fibers in the skin. But how it does this is still unclear.

To find out more, Hofmann et al. studied mutant mice with a disrupted alpha-GAL gene, which consequently lack enzyme activity. Like patients, the mice accumulate Gb3 inside their sensory nerve cells as they age. This build-up of Gb3 damages the cells and reduces the function of ion channels (passages for charged ions to enter and leave a cell) in their membranes. This may contribute to the loss of nerve fibers and the reduced cold-warm sensitivity in Fabry patients.

However, one particular ion channel is more abundant in elderly mutant mice than in normal animals. This channel, called TRPV1, responds to high temperatures and also to capsaicin, the chemical that makes chilli peppers hot. Hofmann et al. propose that the accumulation Gb3 may be linked to the excessive activation of TRPV1 in the sensory nerve cells of patients with Fabry disease. This may in turn contribute to the heat-induced pain.

By providing insights into the mechanisms underlying some of the symptoms of Fabry disease, these findings will assist researchers to develop new treatments. They will also be useful for clinicians who manage patients with the disorder. Further studies should investigate the exact cellular mechanisms linking Gb3 accumulation with changes in cellular activity.

DOI: https://doi.org/10.7554/eLife.39300.002

accumulation might link neuronal pathology with sensory impairment, pain, and peripheral denervation remains to be determined.

We hypothesized that neuronal Gb3 deposits interfere with ion channel expression and function, and neuronal integrity, contributing to the sensory phenotype in FD. We investigated GLA KO mice stratified for age using a comprehensive approach. Our data provide first combined molecular, histological, electrophysiological, and behavioral evidence for a direct and age-dependent influence of intracellular Gb3 deposits on neuronal integrity and ion channel function as a potential mechanism of progressive Fabry-associated sensory disturbance, pain, and skin denervation.

## Results

### Age-dependent Gb3 accumulation in DRG neurons of GLA KO mice is associated with increased endoplasmic stress and skin denervation

First, we examined DRG neuron size by analysing neuronal area (*Figure 1A–D*) and found larger DRG neurons in young GLA KO compared to young WT mice (p<0.01; *Figure 1E*). Neurons of old GLA KO mice were larger compared to old WT (p<0.001) and young GLA KO mice (p<0.001; *Figure 1E*). We also asked if Gb3 deposits are present and where they are located in DRG neurons of young and old GLA KO mice. We assessed semithin sections and found intraneuronal deposits in young and even more so in old GLA KO mice, while DRG neurons from wildtype (WT) mice displayed normal histology (*Figure 1F–I*). We then applied antibodies against CD77 to detect Gb3 and saw marked immunoreaction in DRG of old GLA KO mice, which was not detectable in young mice and in WT littermates (*Figure 1J–M*). Interestingly, Gb3 immunoreactivity was not restricted to neurons, but was also present extra-neurally (*Figure 1M*, arrowheads). Applying confocal microscopy and co-immunoreaction with antibodies against β-(III)-tubulin, we found that Gb3 is mainly located in the cytoplasm of DRG neurons of old GLA KO mice but also in the very proximal parts of sensory axons, in extra-neural connective tissue, and cellular membranes (*Video 1*).

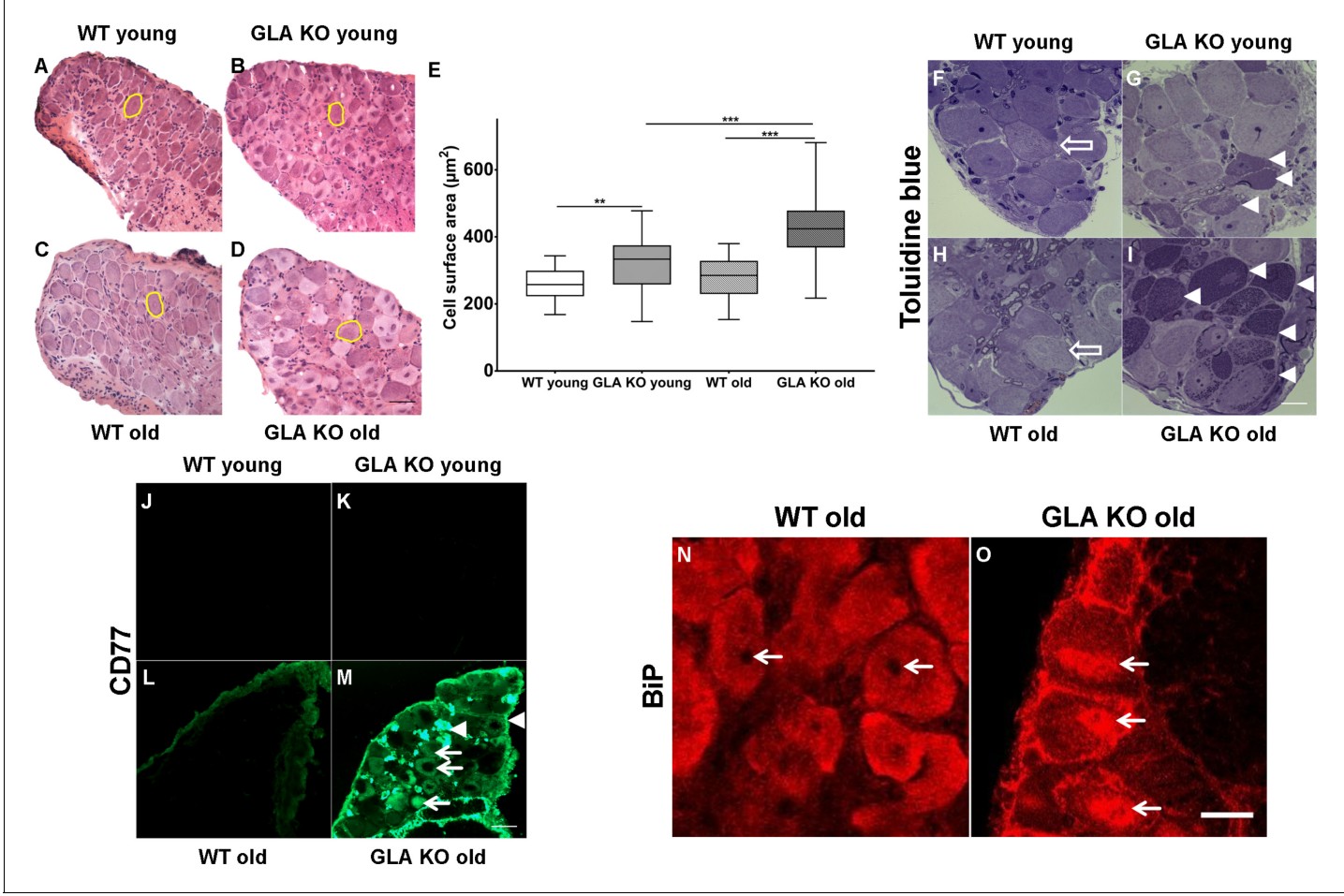

**Figure 1.** Toluidin blue staining and immunoreaction against globotriaosylceramide and immunoglobulin binding protein of mouse dorsal root ganglia. Photomicrographs display hematoxylin-eosin staining DRG neurons from young and old GLA KO and WT mice (A–D) and exemplified measured cell area (yellow circles). (E) Quantification of neuronal cell area revealed increased cell size in young GLA KO compared to young WT mice (p<0.01) and in old GLA KO compared to young GLA KO and old WT mice (p<0.001 each). Photomicrographs show toluidin blue staining (F–I) of 0.5 µm semithin sections of dorsal root ganglia (DRG) from young (3 months) and old (≥12 months) wildtype (WT) and α-galactosidase A deficient (GLA KO) mice. Additionally, photomicrographs display immunoreactivity of antibodies against CD77 as a marker for globotriaosylceramide (Gb3) (J–M) and against binding immunoglobulin protein (BiP) (N–O) on 10 µm cryosections of DRG of old GLA KO and WT mice. No deposits were found in DRG neurons of young WT mice (F, arrow), neurons of a young GLA KO mice showed few intraneuronal deposits (G, arrowheads). Similar to young WT mice, there were no deposits in DRG neurons of old WT mice (H, arrow). Old GLA KO mice, however, displayed many deposits in DRG neurons (I, arrowheads). Gb3 load was not different between young GLA KO, young WT, and old WT mice (J–M), while old GLA KO mice displayed increased Gb3 accumulation in DRG neurons (M, arrows) and extraneural structures (M, arrowheads). BiP was homogeneously expressed in DRG neurons of old WT mice sparing the nucleus (N, arrows). Neurons of old GLA KO mice showed increased accumulation of BiP around the nucleus, indicating accumulation in the endoplasmic reticulum (O, arrows). GLA KO: young (3 months; hematoxylin-eosin: male; toluidine: female; CD77: male), old (≥12 months; hematoxylin-eosin: female; toluidine: female; CD77: male). WT: young (3 months; hematoxylin-eosin: male; toluidine: female; CD77: male), old (≥12 months; hematoxylin-eosin: female; toluidine: male; CD77: male). Scale bar hematoxylin-eosin: 50 µm. Scale bar toloudin blue: 10 µm. Scale bar CD77: 50 µm. The non-parametric Mann-Whitney U test was applied for group comparison. **p<0.01; ***p<0.001.

DOI: https://doi.org/10.7554/eLife.39300.003

To investigate whether Gb3 accumulation in DRG neurons is associated with endoplasmic stress, we performed cellular binding immunoglobulin protein (BiP) expression analysis. BiP was homogeneously distributed in neurons of young GLA KO and WT mice (data not shown) and in old WT mice (*Figure 1N*). In contrast, in neurons of old GLA KO mice, condensed BiP was located within and around the nucleus (*Figure 1O*) indicating enhanced endoplasmic stress.

We then asked, whether increased neuronal Gb3 deposition and endoplasmic stress are associated with a reduction of peripheral innervation, a phenomenon reported for young GLA KO mice

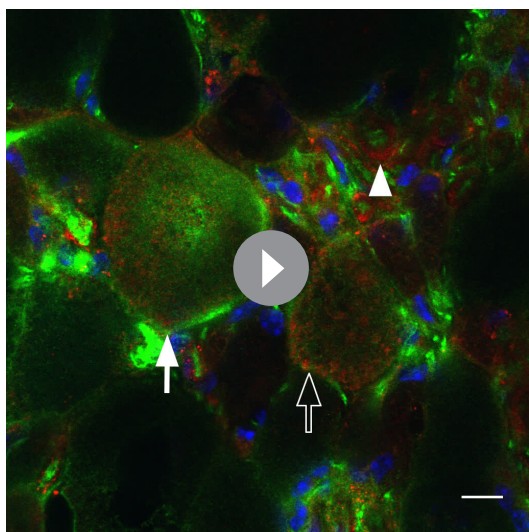

**Video 1.** Localization of globotriaosylceramide in dorsal root ganglion neurons of an old α-galactosidase A deficient mouse Video shows immunoreaction against CD77 (red) as a marker for globotriaosylceramide (Gb3) accumulation and β-(III)-tubulin (green) as a neuron specific cytoplasm marker, in dorsal root ganglion (DRG) neurons of an old (24 months) α-galactosidase A knockout mouse (GLA KO), obtained by confocal laser scanning microscopy. CD77 and β-(III)-tubulin are co-localized during the whole video sequence until the cell body (arrow) is scanned to the middle of the nucleus (end of video), providing evidence that Gb3 deposits (empty arrow) are localized in neuronal cytoplasm, but also in extra-neuronal tissue and proximal parts of axons (arrowhead). Scale bar: 10 μm.
DOI: https://doi.org/10.7554/eLife.39300.004

(*Lakomá et al., 2014*) and known in patients with FD (*Maag et al., 2008*; *Üçeyler et al., 2011*). We quantified intraepidermal nerve fiber density (IENFD) in skin obtained from mouse hind paws and found a marked reduction of cutaneous innervation in young and old GLA KO mice compared to their WT littermates (*Figure 2A–D*), surpassing the physiological reduction of IENFD with aging (p<0.001 each, *Figure 2E*). Furthermore, we assessed whether Gb3 accumulates not only in DRG, but also in axons of the sciatic nerve and in skin. We did not find any Gb3 depositions in the sciatic nerve (*Figure 2F–K*) or footpad skin (*Figure 2L–Q*) of old GLA KO and WT mice.

## Increased apoptosis and decreased neurite outgrowth in cultured DRG neurons of old GLA KO mice

To investigate the degree of apoptosis in DRG neurons in the course of Gb3 accumulation and potential endoplasmic stress, we performed a NucView 488 Caspase 3 Enzyme Substrate Assay. We quantified the percentage of caspase 3 positive neurons in cultured DRG neurons of old GLA KO and WT mice (*Figure 3A–D*). DRG neuron cultures of old GLA KO mice in the naïve state displayed a higher percentage of caspase 3 positive neurons compared to old WT mice (p<0.001, *Figure 3E*) indicating enhanced apoptosis. Additionally, positive control neurons of both genotypes incubated with 500 nM staurosporine for 16 hr showed a higher percentage of caspase 3 positive neurons compared to cultured DRG neurons in the naïve state (p<0.05 each, *Figure 3E*). We further determined the percentage of neurons with neurite outgrowth. Cultured DRG neurons of old GLA KO mice showed less neurite outgrowth compared to neurons of WT mice (p<0.001, *Figure 3F*).

## Increase in TRPV1 protein expression in DRG of old GLA KO mice is associated with enhanced and sustained heat induced pain behavior

Heat intolerance and heat induced pain are key symptoms reported by Fabry patients (*Üçeyler et al., 2014*). We thus investigated transient receptor potential vanilloid 1 (TRPV1) channel expression and function as the major neuronal ion channel that is primarily involved in heat perception and pain. While TRPV1 gene expression did not differ between genotypes and age-groups (*Figure 4A*), we found an increased number of TRPV1 immunoreactive DRG neurons in young and old GLA KO mice compared to their WT littermates (p<0.001 each, *Figure 4B–F*). We also analyzed the distribution of TRPV1 immunoreactivity across different neuronal sizes and quantified TRPV1 positive neuron diameters; neuron populations were stratified as small (<25 μm in diameter) and large (≥25 μm in diameter) neurons (*Figure 4G*)(*Cesare and McNaughton, 1996*; *Hoheisel et al., 1994*; *Lawson et al., 1993*). TRPV1 immunoreactivity was mainly observed in small diameter neurons independent of genotype and age.

Next, we investigated capsaicin induced TRPV1 current densities with patch-clamp analysis in five days old cultured DRG neurons. Neurons appeared enlarged and carried deposits in GLA KO mice, while were of normal shape in WT mice (*Figure 4G,H*). We observed a tendency for higher current

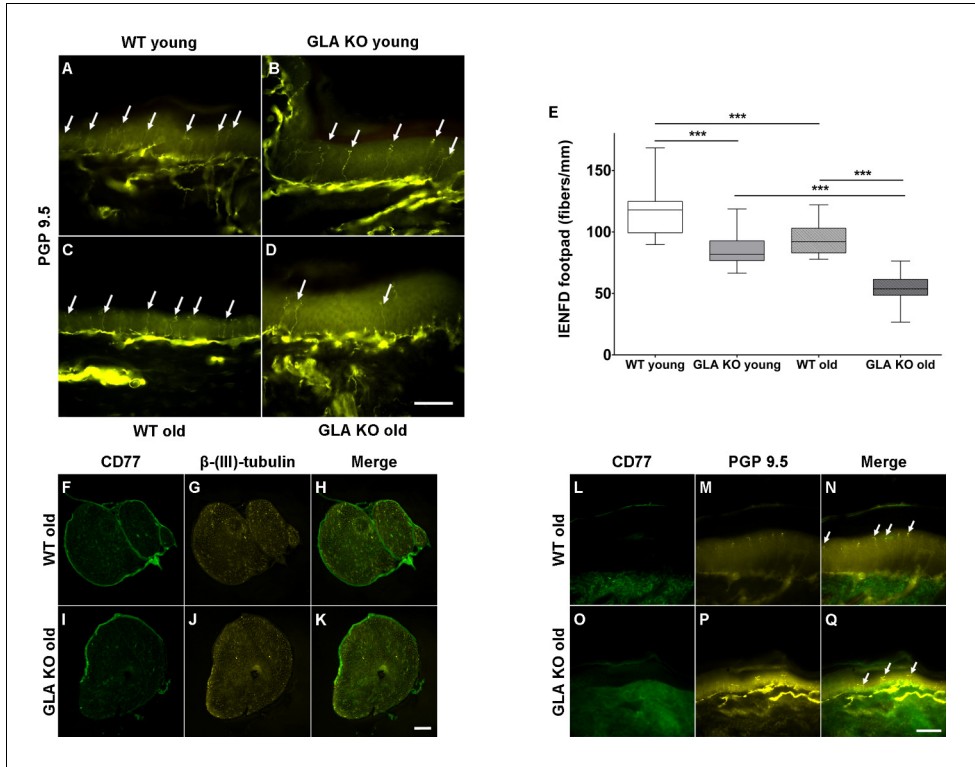

**Figure 2.** Reduced intraepidermal nerve fiber density in α-galactosidase A deficient mice and Gb3 distribution in sciatic nerve and skin. Photomicrographs show immunoreactivity of antibodies against protein gene product 9.5 (PGP 9.5) as a pan-axonal marker in 40 µm skin sections from footpads of young (3 months) and old (≥12 months) wildtype (WT) and α-galactosidase A deficient (GLA KO) mice (**A–D**). Arrows indicate single intraepidermal nerve fibers. Boxplots (**E**) show quantification of intraepidermal nerve fiber density (IENFD). Young WT mice had a higher IENFD compared to young GLA KO and old WT mice (p<0.001, each). Old GLA KO mice showed the most prominent IENFD reduction compared with young GLA KO and old WT mice (p<0.001 each). Additionally, photomicrographs display immunoreactivity of antibodies against CD77 and β-(III)-tubulin in 10 µm sciatic nerve sections (**F–K**) and immunoreactivity of antibodies against CD77 and PGP 9.5 in 40 µm skin section (**L–Q**) of old GLA KO and WT mice. There were no Gb3 depositions detectable. GLA KO: young (3 months, n = 11 male, n = 10 female), old (≥12 months, n = 8 male, n = 11 female). WT: young (3 months, n = 10 male, n = 10 female), old (≥12 months, n = 10 male, n = 9 female). Box plots represent the median value and the upper and lower 25% and 75% quartile. Scale bar: 50 µm. The non-parametric Mann-Whitney U test was applied for group comparison. ***p<0.001.

DOI: https://doi.org/10.7554/eLife.39300.005

densities in young GLA KO mice (exemplified current in *Figure 4I*), but the difference was not significant between genotypes (*Figure 4J*). In contrast, cultured DRG neurons of old GLA KO and littermate WT mice did not respond to capsaicin at all. We investigated neurons obtained from different culture periods (24 hr, three, five, and eight days) so that we do not miss time-dependent TRPV1 currents that might be present only at distinct time points in primary cell culture. TRPV1 currents were also not evoked by capsaicin using calcium-free bath solution to prevent tachyphylaxis. To test for a potential influence of genetic background, we patched DRG neurons of a 14 months old C57BL/6N male mouse, and again did not find capsaicin induced TRPV1 currents under any of the conditions detailed above.

Since increased neuronal TRPV1 protein expression may be associated with heat hypersensitivity, we determined paw withdrawal latencies after intraplantar injection of capsaicin in old GLA KO mice at a dose that induced only mild and short lasting pain behavior in WT mice (*Carey et al., 2017*; *Sakurada et al., 1992*). Indeed, old GLA KO mice showed heat hypersensitivity compared to baseline 24 hr after capsaicin (p<0.01 *Figure 4L*).

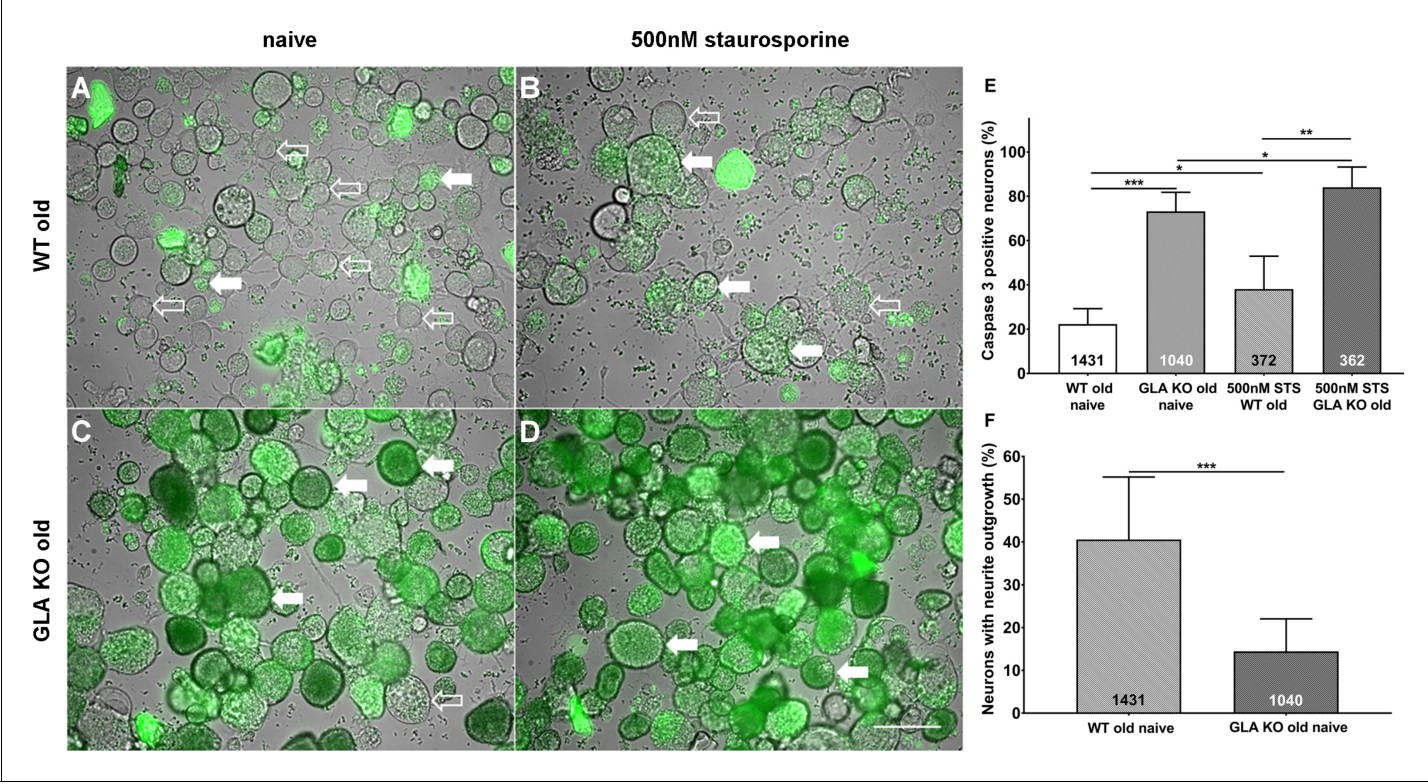

**Figure 3.** More apoptosis and less neurite outgrowth in dorsal root ganglion neurons of old α-galactosidase A deficient mice compared to wildtype mice. Photomicrographs show the results of a NucView 488 Caspase 3 Enzyme Substrate Assay of cultivated dorsal root ganglion (DRG) neurons from old (≥12 months) wildtype (WT) and α-galactosidase A deficient (GLA KO) mice in the naïve state and after incubation with 500 nM staurosporine (STS) as a positive control (A–D). Empty arrows indicate caspase 3 negative neurons and filled arrows point to caspase 3 positive neurons. Bar graphs show the quantification of caspase 3 positive neurons (E). Cultured DRG neurons of old WT mice in the naïve state displayed a lower percentage of caspase 3 positive neurons than those of old GLA KO mice (p<0.001) and neurons of old WT mice incubated with 500 nM STS (p<0.05). DRG neurons of old GLA KO mice incubated with 500 nM STS showed a higher percentage of caspase 3 positive neurons compared to neurons in the naïve state (p<0.05) and WT positive control neurons (p<0.01). Further, neurite outgrowth was quantified (F). DRG neurons of old WT mice in the naïve state displayed a higher percentage of neurons with neurite outgrowth after 48 hr cultivation compared to neurons from old GLA KO mice (p<0.001). NucView 488 Caspase 3 Enzyme Substrate Assay was performed three times on cultures derived from three different mice of each genotype. GLA KO: old (≥12 months, n = 2 male, one female). WT: old (≥12 months, n = 2 male, one female). Number of neurons analyzed are integrated into the corresponding bar. Scale bar: 50 μm. The non-parametric Mann-Whitney U test for group comparisons was applied. *p<0.05;**p<0.01;***p<0.001.

DOI: https://doi.org/10.7554/eLife.39300.006

## Reduction of DRG neuron $I_h$ current densities protects old GLA KO mice from heat and mechanical hypersensitivity after peripheral nerve lesion

We then studied potassium/sodium hyperpolarization-activated cyclic nucleotide-gated ion channels (HCN) and focused on HCN2 as a pacemaker current influencing neuronal action potential frequency and pain in several animal models (*Emery et al., 2012*). There was no intergroup difference for HCN2 gene expression in DRG of GLA KO and WT mice (*Figure 5A*), while HCN2 immunoreactivity increased with age in both genotypes (p<0.05, *Figure 5B–F*). In contrast, patch-clamp analysis of DRG neurons revealed that hyperpolarization-activated ($I_h$) current densities (exemplified current in *Figure 5G*), which are carried by all four isoforms of HCN channels, were markedly reduced in old GLA KO mice compared to old WT mice (p<0.001 each, *Figure 5H*), but did not differ between mice of young age-groups. Lacking a HCN2 specific blocker, further electrophysiological HCN channel subclassfication was not possible.

Since HCN2 conditional knockout mice are protected from heat and mechanical hypersensitivity after peripheral nerve lesion (*Emery et al., 2011*), we applied chronic constriction injury (CCI) at the right sciatic nerve of GLA KO and WT littermates. Indeed, heat hypersensitivity only developed in

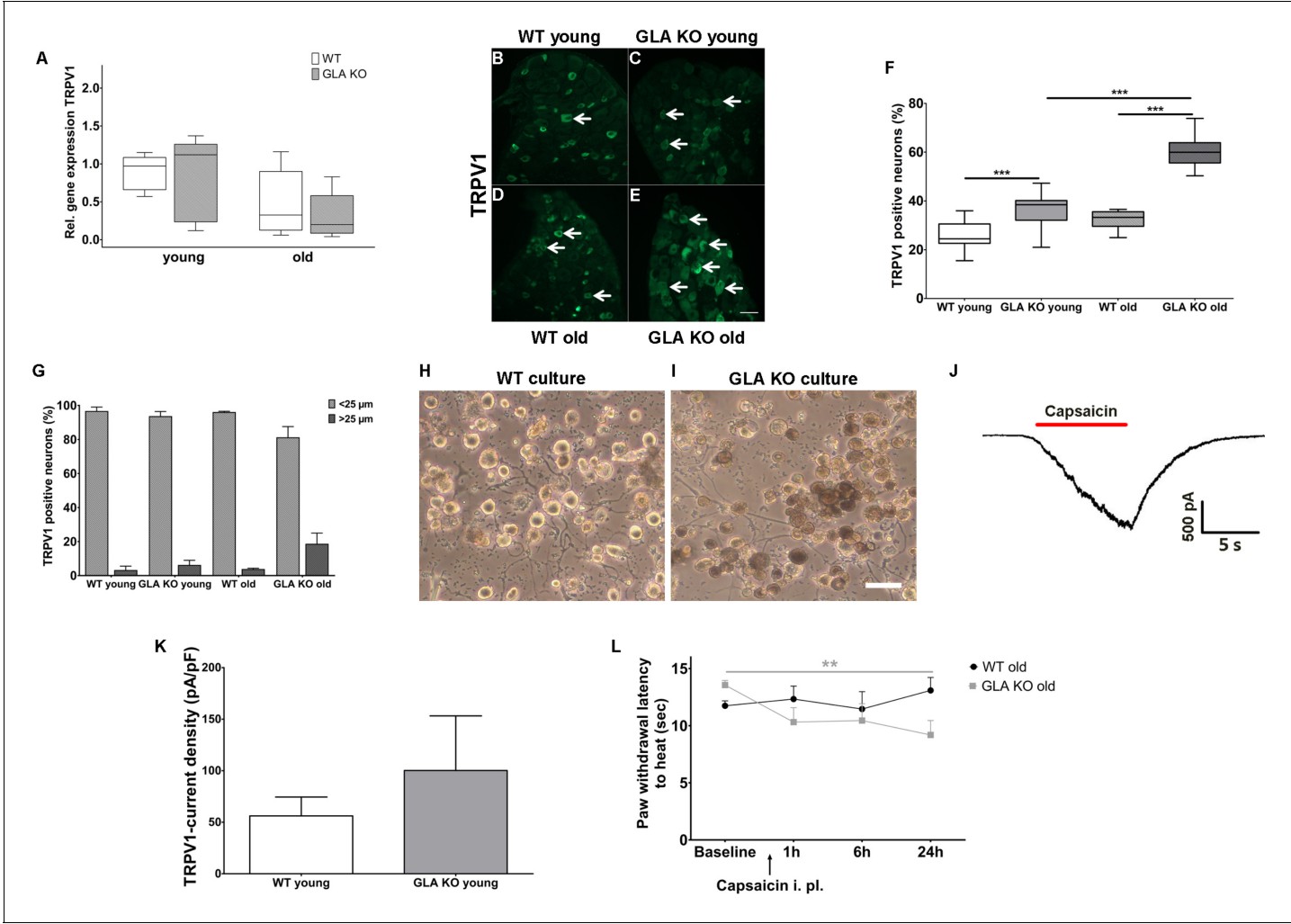

**Figure 4.** Expression, function, and phenotypic reflection of transient receptor potential vanilloid one channels in α-galactosidase A deficient mice. (**A**) Boxplots show the results of transient receptor potential vanilloid 1 (TRPV1) channel gene expression in dorsal root ganglia (DRG) of young (3 months) and old (≥12 months) wildtype (WT) and α-galactosidase A deficient (GLA KO) mice. No intergroup difference was found. (**B–E**) Photomicrographs illustrate immunoreactivity of antibodies against TRPV1 in DRG of young and old WT and GLA KO mice; **F**) shows the result of quantification. Young and old GLA KO mice showed greater TRPV1 immunoreactivity compared to WT littermates (p<0.001 each). (**G**) TRPV1 positive neurons were predominantly smaller than 25 μm in diameter. (**H, I**) Photomicrographs exemplify cultured DRG neurons of an old WT (**H**) and GLA KO mouse (**I**). While cultured neurons appeared normal in WT mice (**H**), intracellular deposits were found in neurons of GLA KO mice (**I**). In J) a TRPV1 current is exemplified which was recorded from cultured DRG neurons of a young GLA KO mouse after application of 500 nM capsaicin (red bar). (**K**) TRPV1 current density did not differ between young GLA KO mice and WT littermates. (**L**) Line charts show thermal withdrawal latencies of old GLA KO and WT mice before and after intraplantar capsaicin injection. Old GLA KO mice displayed increased thermal sensitivty compared to baseline 24 hr after capsaicin injection (p<0.01). GLA KO: young (3 months; gene expression: n = 2 male, n = 4 female; protein expression: n = 12 male, n = 9 female), old (≥12 months; gene expression: n = 4 male, n = 2 female; protein expression: n = 10 male, n = 10 female). WT: young (3 months; gene expression: n = 2 male, n = 4 female; protein expression: n = 9 male, n = 8 female), old (≥12 months; gene expression: n = 3 male, n = 3 female; protein expression: n = 9 male, n = 8 female). TRPV1 currents: Three cells per genotype (GLA KO: n = 1 male, n = 2 female; WT: n = 1 male, n = 2 female) were quantified for calculation of current density. Capsaicin: GLA KO: old (≥12 months; Baseline: n = 33; capsaicin: n = 2 male, n = 6 female). WT old (≥12 months; Baseline: n = 32; capsaicin: n = 3 male, n = 5 female). Behavioral experiments were performed by an observer blinded to the genotype. Scale bar: 50 μm. The non-parametric Mann-Whitney U test for group comparisons was applied. Behavioral data were analyzed using a two-way ANOVA followed by Tukey's post-hoc test.**p<0.01;***p<0.001.

DOI: https://doi.org/10.7554/eLife.39300.007

old WT mice (p<0.01) lasting up to day 28 after surgery, while old GLA KO mice were spared (p<0.01, *Figure 5I*). Also, mechanical withdrawal thresholds remained at baseline level for the entire

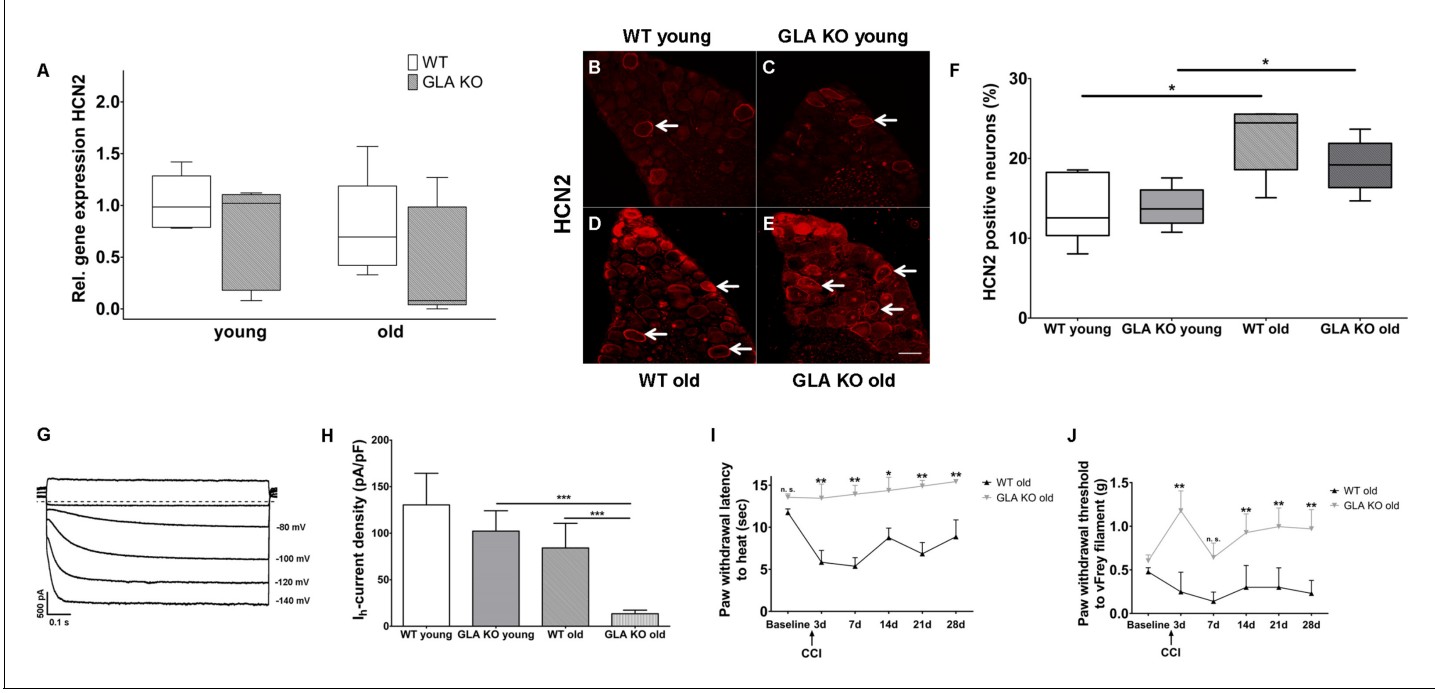

**Figure 5.** Expression, function, and phenotypic reflection of hyperpolarization-activated cyclic nucleotide-gated ion channels in α-galactosidase A deficient mice. (A) Boxplots show the results of potassium/sodium hyperpolarization-activated cyclic nucleotide-gated ion channel 2 (HCN2) gene expression in dorsal root ganglia (DRG) of young (3 months) and old (≥12 months) wildtype (WT) and α-galactosidase A deficient (GLA KO) mice. No intergroup difference was found. (B–E) Photomicrographs illustrate immunoreactivity of antibodies against HCN2 in DRG of young and old WT and GLA KO mice; (F) shows the result of quantification. Old GLA KO and WT mice showed greated HCN2 immunoreactivity compared to young littermates (p<0.05 each) without difference between genotypes. In (G) $I_h$ currents are exemplified recorded from a young GLA KO mouse. (H) $I_h$ current densities did not differ between young GLA KO mice, WT littermates, and old WT mice, while old GLA KO mice displayed markedly reduced $I_h$ current intensities compared to young GLA KO and old WT mice (p<0.001 each). Line charts show thermal withdrawal latencies (I) and mechanial withdrawal thresholds (J) of old GLA KO and WT mice before and after chronic constriction injury (CCI) of the right sciatic nerve. While WT mice showed thermal (p<0.001) and mechanical (p<0.001) hypersensitivity already at day three after CCI lasting up to day 28, old GLA KO mice were spared (p<0.01). GLA KO: young (3 months; gene expression: n = 2 male, n = 4 female; protein expression: n = 2 male, n = 4 female), old (≥12 months; gene expression: n = 4 male, n = 2 female; protein expression: n = 3 male, n = 3 female). WT: young (3 months; gene expression: n = 2 male, n = 4 female; protein expression: n = 4 male, n = 2 female), old (≥12 months; gene expression: n = 3 male, n = 3 female; protein expression: n = 1 male, n = 5 female). $I_h$ currents: At least nine cells per genotype and age-group from at least three different mice each were analyzed. GLA KO young (3 months; n = 4 male, n = 5 female), old (≥12 months; n = 3 male, n = 7 female). WT young (3 months; n = 3 male, n = 6 female), old (≥12 months; n = 4 male, n = 6 female). CCI: GLA KO: old (≥12 months; Baseline: n = 33; CCI: n = 3 male, n = 3 female). WT: old (≥12 months; Baseline: n = 32; CCI: n = 3 male, n = 3 female). Scale bar: 50 µm. The non-parametric Mann-Whitney U test for group comparisons was applied. Behavioral data were analyzed using a two-way ANOVA followed by Tukey's post-hoc test. *p<0.05;**p<0.01;***p<0.001.

DOI: https://doi.org/10.7554/eLife.39300.008

study period of 28 days in old GLA KO mice, while WT mice displayed hypersensitivity to mechanical stimuli up to day 28 after CCI (p<0.01, *Figure 5J*).

## Reduction of DRG neuron Na$_v$1.7 currents protects old GLA KO mice from heat and mechanical hypersensitivity after intraplantar CFA injection

Finally, we investigated neuronal voltage-gated sodium channel 1.7 (Na$_v$1.7) expression and function in GLA KO and WT littermates as a key contributor to neuropathic pain (*Yang et al., 2018*). Na$_v$1.7 gene expression in DRG was not different in both age-groups and genotypes (*Figure 6A*). Since Na$_v$1.7 immunostaining failed using five different Na$_v$1.7 antibodies detailed above, we applied enzyme-linked-immuno-sorbent assay on DRG samples and found no difference in Na$_v$1.7 protein expression in DRG between genotypes and age-groups (*Figure 6B*).

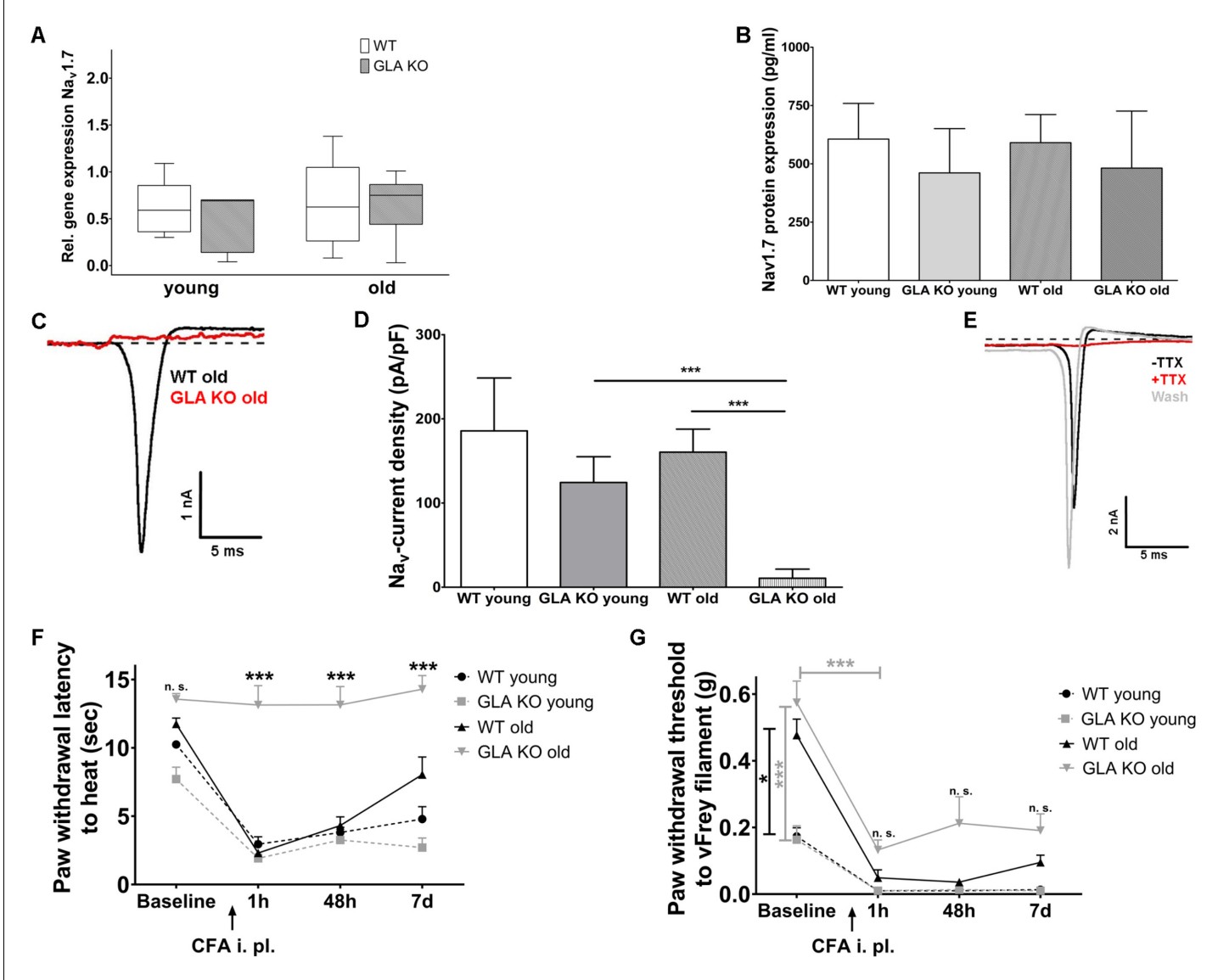

**Figure 6.** Expression, function, and phenotypic reflection of voltage-gated soium channel 1.7 in α-galactosidase A deficient mice. (**A**) Boxplots show the results of voltage-gated sodium channel 1.7 (Na$_v$1.7) gene expression in dorsal root ganglia (DRG) of young (3 months) and old (≥12 months) wildtype (WT) and α-galactosidase A deficient (GLA KO) mice. No intergroup difference was found. (**B**) Na$_v$1.7 protein levels, as investigated by enzyme-linked immunosorbent assay, did not differ between genotypes and age-groups. (**C**) Graphs display exemplified sodium currents of old GLA KO and WT mice at −40 mV. Comparing young GLA KO mice with young and old WT mice, there no difference was found in current densities of sodium currents (**D**), while sodium currents of old GLA KO mice were markedly reduced compared to young GLA KO (p<0.001) and old WT mice (p<0.001). At baseline (**E**, black trace) sodium currents showed fast inactivation kinetics. After adding 100 nM tetrodotoxin (TTX) to the bath solution, sodium currents were completely blocked (**E**, red trace). Sodium currents recovered completely after washout with bath solution (**E**, grey trace). Line charts show thermal withdrawal latencies (**F**) and mechanical withdrawal thresholds (**G**) of young and old GLA KO and WT mice before and after intraplantar injection of complete Freund's adjuvant (CFA). While young and old WT mice showed thermal and mechanical hypersensitivity already one hour after CFA injection lasting up to day seven, old GLA KO mice were spared from heat hypersensitivity (p<0.001). GLA KO: young (3 months; gene expression: n = 2 male, n = 4 female; protein expression: n = 3 male, n = 1 female), old (≥12 months; gene expression: n = 4 male, n = 2 female; protein expression: n = 2 male, n = 2 female). WT: young (3 months; gene expression: n = 2 male, n = 4 female; protein expression: n = 2 male, n = 2 female), old (≥12 months; gene expression: n = 3 male, n = 3 female; protein expression: n = 2 male, n = 2 female). Sodium currents: At least nine cells per genotype and age-group from at least three different mice each were analyzed. GLA KO young (3 months; n = 4 male, n = 5 female), old (≥12 months; n = 3 male, n = 7 female). WT young (3 months; n = 3 male, n = 6 female), old (≥12 months; n = 4 male, n = 6 female). CFA: GLA KO: young (3 months; n = 4 male, n = 2 female), old (≥12 months; Baseline: n = 33; CFA: n = 6 male, n = 6 female). WT: young (3 months; n = 4 male, n = 2 female), old (≥12 months; Baseline 32; CFA: n = 6 male, n = 6 female). Scale bar: 50 μm. The non-parametric Mann-Whitney U test for group comparisons was applied. Behavioral data were analyzed using a two-way ANOVA followed by Tukey's post-hoc test. *p<0.05;***p<0.001.

DOI: https://doi.org/10.7554/eLife.39300.009

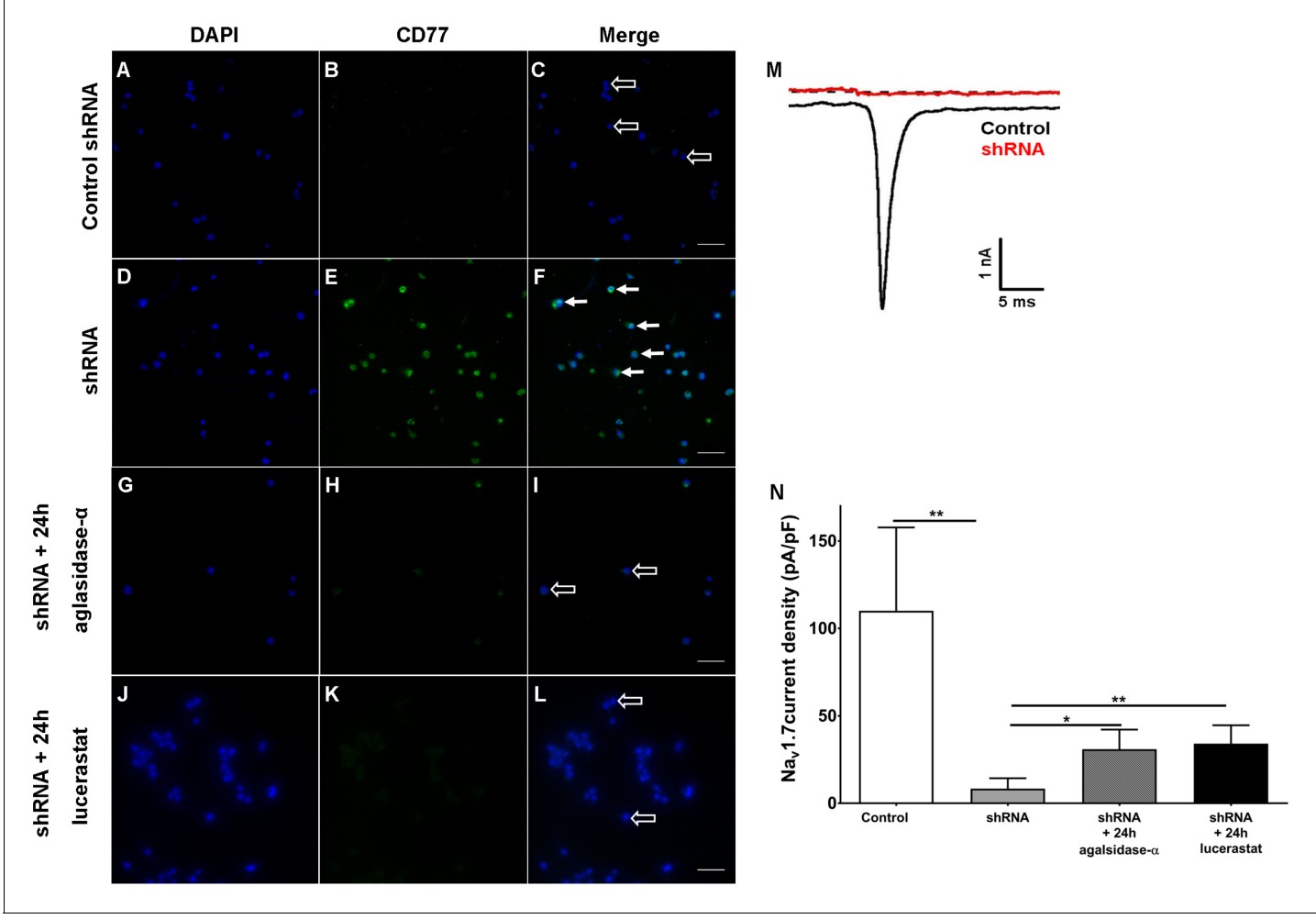

**Figure 7.** Knock-down of α-galactosidase A in human embryonic kidney 293 cells expressing voltage-gated sodium channel 1.7. Photomicrographs show immunoreactivity of antibodies against CD77 as a marker for globotriaosylceramide (Gb3) accumulation in human embryonic kidney 293 (HEK) cells expressing voltage-gated sodium channel 1.7 (Na$_v$1.7) after one week of transfection with control small hairpin RNA (shRNA) (control HEK cells) (**A-C**, empty arrows), shRNA against α-galactosidase A (shRNA HEK cells) (**D-F**, arrows), and after 24 hr of incubation with agalsidase-alpha (**G-I**, empty arrows) and lucerastat (**J-L**, empty arrows). (**M**) Exemplified sodium currents of HEK cells transfected with control shRNA (black) and shRNA (red). (**N**) shRNA HEK cells displayed a marked reduction of Na$_v$1.7 currents compared to control shRNA HEK cells (p<0.01). Treatment with agalsidase-α (p<0.05) and lucerastat (p<0.01) restored Na$_v$1.7 currents. Na$_v$1.7 currents were not different between shRNA treated HEK cells incubated with agalsidase-α, or lucerastat and control cells. Control: n = 16; shRNA: n = 16; shRNA+ 24 hr agalsidase- α: n = 6; lucerastat: n = 11. Bar graphs represent the mean and standard error of the mean and at least three biological replicates. Scale bar 50 μm. The non-parametric Mann-Whitney U test for group comparison was applied. *p<0.05, **p<0.01.

DOI: https://doi.org/10.7554/eLife.39300.010

Patch-clamp analysis revealed that sodium current densities (exemplified currents in *Figure 6C*) were not different between young GLA KO and WT littermates, but were notably reduced in old GLA KO mice compared to old WT mice (p<0.001 each, *Figure 6D*). We applied tetrodotoxin (TTX) to DRG neurons obtained from young GLA KO mice that had normal sodium currents with fast inactivating kinetics at baseline (black trace in *Figure 6E*). These sodium currents were sensitive to TTX already at a concentration of 100 nM (red trace in *Figure 6E*) and recovered after washout with bath solution (grey trace in *Figure 6E*), such that the observed sodium currents were identified as being predominantly produced by Nav1.7, a channel which has been shown to contribute about 70% of the TTX sensitive current in small DRG neurons (*Vasylyev et al., 2014*).

We then investigated whether reduced neuronal Na$_v$1.7 currents may be associated with protection from heat and mechanical hypersensitivity in an inflammatory pain model, as known for Na$_v$1.7

conditional knockout mice (*Nassar et al., 2004*). Indeed, intraplantar injection of complete Freund's adjuvant (CFA) led to heat hypersensitivity in all mice groups except for old GLA KO mice (p<0.001, *Figure 6F*), in which heat withdrawal latencies did not change from baseline for the entire study period of seven days (p<0.001, *Figure 6F*). Similarly, all mice developed mechanical hypersensitivity starting one hour after CFA injection compared to baseline (p<0.001, *Figure 6G*), which was less pronounced in old GLA KO mice compared to old WT mice after CFA injection (*Figure 6G*), and all mice remained mechanically hypersensitive until day seven after CFA injection.

## Gb3 accumulation and reversible reduction of Na$_v$1.7 currents in HEK cells after shRNA treatment

Finally, we investigated if cellular Gb3 accumulation interferes with Na$_v$1.7 currents. For this, we silenced α-GAL in human embryonic kidney 293 cells (HEK) expressing Na$_v$1.7 with small hairpin RNA (shRNA) directed against the human α-GAL transcript as an in vitro model. Few HEK cells transfected with control shRNA (control HEK cells, *Figure 7A–C*) showed mild Gb3 deposition, while the majority of HEK cells transfected with shRNA against α-GAL (shRNA HEK cells, *Figure 7D–F*) displayed a marked increase in Gb3 accumulation within only one week of transfection. These Gb3 deposits were reversible by incubation with 1.32 μg/ml agalsidase-α (1 mg/ml, Shire, Saint Helier, Jersey, UK) and 250 μM lucerastat (N-butyldeoxygalactonojirimycin, Biomol,cat# Cay19520-1, Hamburg, Germany) applied for 24 hr prior to patch-clamp recordings (*Figure 7G–L*). Electrophysiological analysis of Na$_v$1.7 currents in Gb3-positive HEK cells revealed a marked decrease of sodium currents after shRNA treatment compared to control HEK cells (p<0.01, *Figure 7J,K*), which recovered after agalsidase-α and lucerastat incubation (agalsidase-α: p<0.05; lucerastat: p<0.01, *Figure 7N*).

## Discussion

We comprehensively investigated the impact of sensory neuron Gb3 deposits in the α-GAL deficient mouse model as a potential basis of small fiber neuropathy in FD and detected three major effects: Gb3 is age-dependently associated with (1) increased BiP expression indicating endoplasmic stress and nerve fiber degeneration, (2) increased neuronal TRPV1 protein expression and sustained capsaicin responsiveness in vivo, and (3) reduced neuronal I$_h$ and Na$_v$1.7 currents associated with a lack of thermal and mechanical hypersensitivity after nerve lesion and inflammation.

Early autopsy reports pointed to potential neuronal Gb3 deposits (*Gadoth and Sandbank, 1983*; *Kaye et al., 1988*), which we also found in DRG neurons of young GLA KO mice (*Lakomá et al., 2016*; *Namer et al., 2017*). We assessed Gb3 deposits in DRG of young and old GLA KO mice and show age-dependent intra- but also extra-cellular Gb3 accumulation challenging the concept of exclusive lysosomal storage. We hypothesize that exceeding compensation limits, Gb3 deposits might break loose from lysosomes getting into contact with other organelles and cellular structures. Alternatively, Gb3 may be produced and secreted by surrounding non-neuronal cells. This may underlie Gb3 associated cellular stress and apoptosis as shown for example in cardiomyocytes (*Chimenti et al., 2015*), peripheral blood mononuclear cells (*De Francesco et al., 2011*) or endothelial cells (*Shen et al., 2008*) of patients with FD. Endoplasmic stress, as found in DRG neurons of old GLA KO mice (*Figure 1*), is a major trigger of apoptosis (*Wang et al., 2009*), which may be the basis of Gb3-dependent skin denervation as a hallmark of FD (*Maag et al., 2008*; *Üçeyler et al., 2011*). Indeed, DRG neurons of old GLA KO mice also displayed increased caspase three activity and decreased neurite outgrowth as markers of apoptosis. Increased caspase three activity is associated with cellular vulnerability and apoptotic cell death (*Hartmann et al., 2000*) and is involved in DNA breakdown and morphological changes during apoptosis (*Jänicke et al., 1998*).

Alterations of neuronal ion channel expression and function have long been assumed to be potential contributors to sensory impairment and pain in FD. Higher nociceptor TRPV1 expression was reported in young GLA KO mice compared to WT mice with a mild and transient increase in TRPV1 currents of DRG neurons upon high-dose capsaicin treatment in vitro and heat intolerance in the hot plate test (*Lakomá et al., 2016*). We recently showed heat hypersensitivity in naïve young GLA KO mice also in the Hargreaves test, which turned to hyposensitivity with aging (*Üçeyler et al., 2016*). Adding to this evidence, we here report on higher TRPV1 protein immunoreactivity in DRG neurons of young and old GLA KO mice compared to WT littermates without changes in gene

expression and show that old GLA KO mice develop sustained heat hypersensitivity when treated with capsaicin. Thus, increased neuronal TRPV1 protein immunoreactivity may contribute to heat hypersensitivity in naïve young GLA KO mice (*Lakomá et al., 2016*; *Üçeyler et al., 2016*) and may turn to heat hyposensitivity with aging (*Üçeyler et al., 2016*) due to stress-induced degeneration of peripheral afferents. However, challenging the system by capsaicin may still induce heat hypersensitivity despite skin denervation due to the high expression of neuronal TRPV1 channels as shown for old GLA KO mice here. It remains unclear though, if the increase in TRPV1 protein immunoreactivity and the capsaicin-induced heat hypersensitivity is also associated with neuronal TRPV1 channel dysfunction. It is of note that acute heat sensitivity is based on three different transient receptor potential channels indicating high redundancy (*Vandewauw et al., 2018*). A recent study investigating a rat model of FD provided evidence for TRPA1 dependent mechanical but not thermal hypersensitivity in a Fabry rat model without differences in TRPV1 currents in young rats (*Miller et al., 2018*). In line with these results, current properties of TRPV1 did not differ between young GLA KO and WT mice in our experiments (*Figure 3J*). Extensive patch-clamp analysis of neurons obtained from old mice did not reveal capsaicin induced currents at all. Since TRPV1 currents upon capsaicin stimulation were also absent in old littermate WT and C57BL/6N mice, we assume this to be a physiological age-dependent finding.

All four HCN channel isoforms are expressed in DRG neurons and contribute to neuronal excitability and generation of action potential rhythmicity (*Biel et al., 2009*). Since the first description of HCN2 as a pacemaker channel under inflammatory and neuropathic pain conditions (*Emery et al., 2011*), several studies were performed confirming and extending these data and mainly showing heat and mechanical hypersensitivity upon HCN2 channel activation (*Herrmann et al., 2017*; *Schnorr et al., 2014*; *Tsantoulas et al., 2017*). We confirm previous findings of an age-dependent increase of DRG neuron HCN2 protein expression (*Hou et al., 2015*) independent of genotypes. Despite this increase, old GLA KO mice showed a marked decrease of $I_h$ currents when compared with WT littermates. A converse relation between protein expression and $I_h$ current density was described previously (*Chaplan et al., 2003*). The underlying mechanism of this discrepancy remains unclear. One possibility is that despite an increase of the HCN2 isoform, other HCN channels might be downregulated resulting in low overall $I_h$ currents. Although our behavioral data mimicking findings in HCN2 conditional knockout mice (*Emery et al., 2011*) suggest that HCN2 may be the underlying major channel, we cannot rule out the contribution of other HCN isoforms lacking an HCN2 specific blocker. Furthermore, $I_h$ currents are mixed and mainly based on sodium and potassium flux (*Biel et al., 2009*). Thus, the reduction in sodium current densities as shown in our study may also cross-contribute to the observed drop in $I_h$ current densities in old GLA KO mice.

Voltage-gated sodium channels are key contributors to small fiber associated pain syndromes (*Waxman et al., 2014*), however, data on sodium channel expression and functionality in GLA KO mice are scarce. In one study, young GLA KO mice displayed increased expression of skin $Na_v1.8$ protein, while $Na_v1.7$ expression did not differ between genotypes (*Lakomá et al., 2014*). In another study not reporting the age of experimental animals, a generally decreased conductance of sodium currents was described in DRG neurons of GLA KO compared to WT mice (*Namer et al., 2017*). In a recent study investigating potential gene expression signatures in DRG of GLA KO mice a slight downregulation of sodium channel protein type seven subunit alpha was reported compared to WT mice as the only finding in sodium channel genes (*Kummer et al., 2017*). While we did not find differences in the gene and protein expression of $Na_v1.7$ in DRG neurons comparing GLA KO and WT mice, we provide first evidence for the age-dependent dampening of neuronal $Na_v1.7$ currents in old GLA KO mice. Most intriguingly, this reduction of $Na_v1.7$ currents was reproducible in vitro by silencing α-GAL in HEK cells and was reversed by treatment of HEK cells with the Fabry-specific drug used as enzyme replacement therapy in Fabry patients, agalsidase-α and a new Gb3 synthase inhibitor lucerastat (*Figure 7*), pointing to a functional effect of Gb3 on $Na_v1.7$ channels.

We provide first evidence for a direct link between intracellular Gb3 deposits and neuronal ion channel dysfunction as the potential basis of Fabry associated sensory symptoms and signs. The question remains about the exact mechanism underlying Gb3 mediated channel dysfunction. One possibility is the direct interaction of extra-lysosomal Gb3 with membrane structures such as lipid rafts anchoring ion channels (*Schnaar, 2016*; *Sural-Fehr and Bongarzone, 2016*). Disturbance of cell membrane homeostasis by intracellular Gb3 deposits may alter channel function independent of gene or protein expression. Another question is, if and how changes in functionality of one channel

family may influence other neuronal ion channels and if cross-communication may underlie some of the effects observed here. We can also not rule out the effect of further ion channels such as potassium or calcium that have been reported to be potentially affected by Gb3 in different experimental settings. For instance, calcium dependent potassium channel type 3.1 was age-dependently reduced in aortic endothelial cells of GLA KO mice (*Park et al., 2011*). In turn, Gb3 enhanced voltage-gated calcium currents of sensory DRG neurons in vitro and led to mechanical allodynia after intraplantar injection in WT mice (*Choi et al., 2015*). Thus, intracellular Gb3 deposits may exert effects on membrane ion channels in general and disturb their functional composition leading to sensory symptoms and pain.

## Conclusions

Our data give first evidence for the involvement of neuronal Gb3 deposits in the pathophysiology of skin denervation and a direct and major role in sensory impairment, and pain of patients with FD. The exact mechanisms, however, remain to be elucidated, we show that neuronal Gb3 deposits result in an overall reduction of ion channel current densities and provide a HEK cell based in vitro model as a potent tool for further pathophysiological research and pharmaceutical testing of new Fabry-specific drugs. Gb3 influences neuronal function and integrity, thus, a sustained normalization of intracellular Gb3 load by drugs providing permanently low Gb3 levels without recurrent end-of-dose peaks is crucial which may be achieved with new pharmaceutical formulations. Our study also underscores the importance of investigating further neuronal ion channels like $Na_v$ and HCN isotypes and of studies in other organ systems, such as the heart and kidneys, to better understand the effect of Gb3 on for example cardiomyocytes in the generation of lethal arrhythmias. We believe that such approaches will open new avenues for mechanism-based diagnostics and treatment options for patients suffering from the life threatening FD.

# Materials and methods

## Mice and study groups

Our study was approved by the Bavarian State authorities (Regierung von Unterfranken, # 54/12). Animal use and care was in accordance with institutional guidelines. Mice were held in the animal facilities of the Department of Neurology, University of Würzburg, Germany. They were fed standard chow (commercially prepared complete diet) and had food and water access ad libitum.

We used 95 GLA KO mice (45 male, 50 female) of mixed genetic background (C57BL6 and SVJ129) carrying a targeted disruption of the α-galactosidase A gene (*GLA*) as previously described (*Ohshima et al., 1997*). Additionally, 96 WT littermate mice (45 male, 51 female) were assessed. To ensure that our KO and WT mice have an identical genetic background, we first crossed GLA KO mice with C57BL6/N mice to generate heterozygous off-springs. These heterozygous mice were then cross-bred with each other to obtain homozygous female and male GLA KO and WT mice. In the further course of breeding, we mated these two homozygous lines only with genetically matching mice (KO x KO, WT x WT) of the respective strain. Thus, we generated two mouse strains (homozygous GLA KO and WT) with an identical genetic background. To maintain the purity of these strains, we performed genotype analysis on every single mouse born in our animal facility. For genotyping, we used the Kapa2G fast PCR Kit (Kapa Biosystems, Wilmington, USA) and the following primers: oIMR5947, AGGTCCACAGCAAAGGATTG; oIMR5948, GCAAGTTGCCCTCTGACTTC; oIMR7415, GCCAGAGGCCACTTGTGTAG. Since FD shows age-dependent progression, we investigated young (3 months) and old (≥12 months) mice; old mice reached an age of up to 24 months. Animals were not stratified for sex, since in GLA KO mice α-GAL knockout results in a complete loss of enzyme activity in both sexes (i.e. homozygous female mice, hemizygous male mice) with similar pain behavior in contrast to human patients (*Lakomá et al., 2014*; *Lakomá et al., 2016*; *Rodrigues et al., 2009*; *Üçeyler et al., 2016*).

## Tissue collection

Mice were sacrificed in deep isoflurane anesthesia (CP-Pharma, Burgdorf, Germany) and lumbar L3 and L5 DRG were dissected for quantitative real-time PCR. Tissue was obtained in the naïve state and was flash-frozen in liquid nitrogen for storage at −80°C before further processing. L4 DRG were

collected for immunohistochemistry (see below) and were embedded in optimal cutting temperature medium (TissueTek, Sakura Finetek, Staufen, Germany); ganglia were stored at −80°C before further processing. For neuronal cell cultures, ten to twelve DRG pairs were dissected within 30 min after mice were sacrificed. Skin of footpads was dissected and incubated in 4% paraformaldehyde (PFA, Merck Millipore, cat# 1.04005, Billerica, Massachusetts, USA) for three hours. After washing three times with phosphate buffer, skin samples were incubated in 10% sucrose at 4°C, were embedded in optimal cutting temperature medium, and stored at −80°C before further processing.

## Immunohistochemistry

Right L4 DRG of young and old GLA KO and WT mice were collected in 4% PFA (Merck Millipore, cat# 1.04005; Billerica, Massachusetts, USA) in 2% glutaraldehyde (25% stock solution, Serva, cat# 23115, Heidelberg, Germany). Briefly, tissue was postfixed with 2% osmiumtetraoxid (Chempur, cat# 006051, Karlsruhe, Germany) and dehydrated with an ascending aceton row (Sigma-Aldrich, cat# 15364-56-4, Taufkirchen, Germany). After embedding in plastic, 0.5 µm semithin sections were prepared using an ultramicrotome (Leica EM UC7, Leica Microsystems, Wetzlar, Germany) and were stained with toluidine blue for light microscopy (Axiophot two microscope, Zeiss, Oberkochen, Germany).

Ten-µm DRG and sciatic cryosections were prepared with a cryostat (Leica, Bensheim, Germany). We performed hematoxylin-eosin staining. Briefly, DRG cryosections were incubated in hematoxylin (Sigma-Aldrich, cat# H3136, Taufkirchen, Germany) for 10 min and 25 s with 1% eosin (Sigma-Aldrich, cat# 23251, Taufkirchen, Germany). Afterwards, cryosections were dehydrated with an ascending ethanol row. To quantify cell size, neurons were surrounded using Fiji software (ImageJ 1.50 g, Wayne Rasband, National Institute of Health, USA) (*Schindelin et al., 2012*) and perimeter was calculated. For immunofluorescence, antibodies against TRPV1 (goat, 1:500, Santa Cruz, cat# SC-12498; Santa Cruz, California, USA), and HCN2 (rabbit, 1:200, Alomone Labs, cat# APC-030; Jerusalem, Israel) were used. Five different $Na_v1.7$ polyclonal antibodies were tested (anti-rabbit, Alomone Labs: cat# ASC-008; anti- rabbit, cat# ASC-027; anti-guinea pig, cat# AGP-057, Jerusalem, Israel; anti-mouse, Abcam, cat# ab85015, Cambridge, UK; rabbit anti-$Na_v1.7$: Y083, generated from rat a.a. sequence 514–532, Center for Neuroscience and Regeneration Research, Yale Medical School and Veterans Affairs Hospital, West Haven, Connecticut, USA). Additionally, antibodies against β-(III)-tubulin (chicken, 1:500, Abcam, cat# ab41489, Cambridge, UK), BiP (rabbit, 1:5000, Abcam, cat# ab21685, Cambridge, UK) and CD77 (i.e. Gb3, rat, 1:250, Bio-Rad, cat# MCA579; Hercules, California, USA) were used to document endoplasmic stress responses under pathophysiological conditions (*Lee, 2005*). We used goat anti-rabbit IgG, rabbit anti-goat IgG and goat anti-chicken IgG labelled with cyanine 3.18 fluorescent probe (1:50, Amersham; Piscataway, New Jersey, USA) and Alexa Fluor 488 anti-rat IgM (1:300; Jackson Laboratory; Bar Habor, Maine, USA) as secondary antibodies. Negative controls were prepared by omitting the primary antibody. Photomicrographs were assessed manually (Axiophot two microscope, Zeiss, Oberkochen, Germany) using Spot Advanced Software (Windows Version 5.2, Diagnostic Instruments, Inc, Sterling Heights, USA). For quantification of ion channel positive cells, the total number of neurons per DRG sections (three sections per mouse) were counted with Fiji software (ImageJ 1.50 g, Wayne Rasband, National Institute of Health, USA) (*Schindelin et al., 2012*) and the percentage of immunoreactive neurons relative to the total number of neurons with a clear nucleus was calculated by an observer blinded to the genotype. Additionally, diameter of TRPV1 positive neurons were measured with Fiji software (ImageJ 1.50 g, Wayne Rasband, National Institute of Health, USA) (*Schindelin et al., 2012*) and neurons were categorized into small (<25 µm) and large (>25 µm) neurons.

Forty-µm skin sections from footpads were prepared with a cryostat (Leica, Bensheim, Germany). For immunofluorescence, antibodies against protein gene product-9.5 (PGP9.5, rabbit, 1:500, Ultra-Clone Limited, Isle of Wight, England) were used. We applied goat anti-rabbit IgG labelled with cyanine 3.18 fluorescent probe (1:50, Amersham; Piscataway, New Jersey, USA) as secondary antibody. Intraepidermal nerve fibers were counted and the number of fibers per millimeter was calculated applying published counting rules (*Lauria et al., 2005*).

## Confocal laser scanning microscopy

Confocal microscopy was performed on 10 µm cryosections of DRG obtained as described above. For immunofluorescence, antibodies against CD77 (i.e. Gb3, rat, 1:250, Bio-Rad, cat# MCA579; Hercules, California, USA) and β-(III)-tubulin (chicken, 1:500, Abcam, cat# ab41489, Cambridge, UK) were used. We applied rabbit anti-rat IgM labelled with cyanine 3.18 fluorescent probe (1:50, Amersham; Piscataway, New Jersey, USA) and Alexa Fluor 488 coupled anti-chicken (1:300; Jackson Laboratory; Bar Habor, Maine, USA) as secondary antibodies together with 4',6-diamidino-2-phenylindole (1:10.000; Sigma-Aldrich, cat# 28718-90-3, Taufkirchen, Germany). Photomicrographs were acquired using an inverted IX81 microscope (Olympus, Tokyo, Japan) equipped with an Olympus FV1000 confocal laser scanning system, a FVD10 SPD spectral detector and diode lasers of 405, 473, 559, and 635 nm. All images shown were acquired with an Olympus UPLSAPO60x (oil, numerical aperture: 1.35) objective. For high-resolution confocal scanning, a pinhole setting representing one Airy disc was used. High-resolution confocal settings were chosen to meet an optimum resolution of at least three pixels per feature in x–y direction. In z-direction, 600 nm steps were used. 12-bit z-stack images were processed by maximum intensity projection and were adjusted in brightness and contrast. Images are shown as red-green-blue images. Image and video processing was performed with Fiji (ImageJ 1.50 g, Wayne Rasband, National Institute of Health, USA) (*Schindelin et al., 2012*).

## Gene expression analysis

Frozen DRG samples were processed using a Polytron PT 3100 homogenizer (Kinematica, Luzern, Switzerland). Total RNA was isolated using TRIzol reagent (Invitrogen, Carlsbad, CA, USA) following the manufacturer's instructions. Five hundred ng of RNA were then reverse transcribed with TaqMan Reverse Transcription Reagents (Applied Biosystems, Darmstadt, Germany). Five µl of cDNA per sample were assessed with quantitative real-time PCR using TaqMan Universal Master Mix and the following target specific predesigned mouse TaqMan Gene Expression Assays (Applied Biosystems, Darmstadt, Germany; Assay-IDs in brackets): TRPV1 (Mm01246302_m1), HCN2 (Mm00468538_m1), $Na_v1.7$ (Mm00450762_s1). 18 s rRNA (Hs99999901_s1) was used as an endogenous control. Quantitative real-time PCR reactions were performed in the 96-well GeneAmp PCR System 9700 cycler with the following cycler conditions: 2 min, 50°C; 10 min, 95°C; (15 s, 95°C; 1 min, 60°C) x40. Relative gene expression was calculated using the $2^{-\Delta\Delta Ct}$ method.

## DRG protein analysis

For protein analysis, ten to twelve DRG pairs per mouse were dissected (see above) and frozen at −80°C until further processing. To achieve sufficient tissue weight (i.e. ≥300 mg), DRG of at least three mice were pooled on ice and were processed using a Polytron PT 3100 homogenizer (Kinematica, Luzern, Switzerland) in 500 µl phosphate buffered saline containing 20 µl protease inhibitor. The suspension was centrifuged 15 min at 1500 g and the supernatant was separated in aliquots à 200 µl. A mouse Nav1.7 enzyme-linked immunosorbent assay kit (BlueGene, 0,1 ng/ml, cat# E03N0034, Shanghai, China) was used to determine $Na_v1.7$ protein expression together with provided standards, following the manufacturer's instructions and using undiluted samples.

## DRG neuron cell culture

Mouse DRG neurons were dissected and cultivated in culture medium (100 ml TNB-100, Biochrom, cat# F8023; Berlin, Germany, 25 mM glucose; 2 ml PenStrep, Life Technologies, cat# 15140–122; Carlsbad, CA, USA; 100 µl L-glutamine, Life Technologies, cat# 25030–032; Carlsbad, CA, USA; 2 ml protein-lipid-complex, Biochrom, cat# F8820; Berlin, Germany) containing 25 ng/ml nerve growth factor (2.5S, Alomone Labs, cat# N-240; Jerusalem, Israel) according to a previously published protocol (*Langeslag et al., 2014*).

## Caspase 3 substrate assay

DRG neurons of old GLA KO and WT mice, were dissected and cultured for 48 hr as described above. To analyze apoptosis, we performed a NucView 488 Caspase 3 Enzyme Substrate Assay (Biotium, cat# 10403, Fenton, California, USA) according to the manufacturer's protocol. As a positive control, cells of both genotypes were incubated with 500 nM staurosporine (Abcam, cat# ab120056, Cambridge, UK) for 16 hr prior to performing the NucView 488 Caspase3 Enzyme Substrate Assay.

For quantification of apoptosis, the percentage of caspase three positive neurons and the percentage of neurons with neurite outgrowth was determined.

## Patch-clamp analysis

Whole-cell recordings were performed at room temperature three to eight days after isolation of DRG neurons and after axonal outgrowth. Bath solution consisted of 135 mM NaCl, 5.4 mM KCl, 1.8 mM CaCl$_2$, 1 mM MgCl$_2$, 10 mM glucose, and 5 mM HEPES (*Eberhardt et al., 2017*; *Hamill et al., 1981*). Bath solution for HEK cells consisted of 140 mM NaCl, 3 mM KCl, 1 mM CaCl$_2$, 1 mM MgCl$_2$, and 10 mM HEPES. Patch pipettes were pulled from borosilicate glass capillaries (Kimble Chase Life Science and Research Products, Meiningen, Germany) and were heat-polished to reach an input resistance of 2 to 3 M$\Omega$ (whole-cell). The pipette recording solution contained 140 mM KCl, 2 mM MgCl$_2$, 1 mM EGTA, 1 mM ATP, and 5 mM HEPES for DRG neuron analysis and 140 mM CsF, 1 mM EGTA, 10 mM NaCl, and 10 mM HEPES for HEK cell analysis. Currents were recorded with an EPC9 patch-clamp amplifier (HEKA, Ludwigshafen, Germany) with a sampling rate of 20 kHz. Stimulation and data acquisition were controlled by the PULSE/PULSEFIT software package (HEKA, Lambrecht, Germany) on a Macintosh computer, and data analysis was performed off-line with IGOR software (WaveMetrics, Lake Oswego, Oregon, USA).

To quantify TRPV1 currents, 500 nM capsaicin (Merck Millipore, cat# 21127, Billerica, Massachusetts, USA) was used on DRG neurons. To investigate I$_h$ currents, we used a series of depolarizing and hyperpolarizing step voltage pulses. To identify sodium channels, TTX (Alomone Labs, cat# T-550; Jerusalem, Israel) was applied to DRG neurons at a concentration of 100 nM and 1 µM using a standard perfusion system (Solution Exchange System ALAVC3-8, ALA Scientific Instruments, Farmingdale, New York, USA). Sodium currents were recorded continuously. For the quantification of TRPV1 and sodium currents, we performed measurements at maximum potential amplitudes; I$_h$ currents were recorded at −120 mV. Current density was calculated by normalizing the measured potentials to cell size. DRG neurons with less than 25 pF capacity were considered as nociceptors.

## Treatment and surgery

We investigated the effect of intraplantar injection of one µg capsaicin in 10 µl normal saline (Merck Millipore, Billerica, Massachusetts, USA) to the right hind paw of old GLA KO and WT mice under isoflurane narcosis. In a previous study a comparable dosage of intraplantar capsaicin led to short lasting (<10 min) pain behavior in mice (*Carey et al., 2017*; *Sakurada et al., 1992*). We determined heat withdrawal latencies one, six, and 24 hr after capsaicin injection in old GLA KO and WT mice.

To model neuropathic pain, old mice of both genotypes received CCI of the right sciatic nerve (*Bennett and Xie, 1988*; *Sommer and Schäfers, 1998*). Briefly, mice were anesthetized with isoflurane and the right sciatic nerve was exposed. Three ligatures (7–0 prolene, Ethicon, Norderstedt, Germany) with a distance of one mm each were loosely tied around the nerve proximal to its trifurcation until the ipsilateral hind paw flicked shortly. Behavioral tests were performed at baseline, three, seven, 14, 21, and 28 days after CCI.

To induce inflammatory pain, mice of both genotypes and age-groups received an intraplantar injection of CFA (Sigma-Aldrich, Taufkirchen, Germany). Ten µl CFA (concentration: 20 pg/µl) were applied intraplantarly to the right hind paw under isoflurane anesthesia. Behavioral tests were performed at baseline, one and 48 hr, and seven days after CFA injection. As a control, ten µl of normal saline 0.9% (Braun, Melsungen, Germany) were injected into the right hind paw of each control mouse.

## Behavioral tests

All behavioral tests were performed by the same experienced investigator blinded to the genotype and treatment groups. All animals were examined three times, each with a test interval of 1–2 days before interventions.

Heat withdrawal latencies were determined using the Hargreaves method with a standard Ugo Basile algometer (Comerio, Italy) (*Hargreaves et al., 1988*). Mice were placed on a glass surface within acrylic glass boxes and a radiant heat stimulus (25 IR) was positioned under the plantar surface of the hind paw after 60 min of adaptation. The paw withdrawal latency was measured automatically.

To avoid burn lesions, a stimulus cut-off time of 16 s was set. Each hind paw was tested three times (at intervals of 5 min).

Mechanical withdrawal thresholds were determined using the von-Frey test based on the up-and-down-method (*Chaplan et al., 1994*). Mice within acrylic glass boxes were placed on a wire mesh. After adaption for 60 min, the plantar surface of the hind paw was touched with a von-Frey filament (beginning at 0.69 g). Upon paw withdrawal the next thinner von-Frey filament was applied. If no paw withdrawal was observed, the next thicker von-Frey filament was used. Each hind paw was tested six times (at intervals of 5 min). The 50% withdrawal threshold (i.e. force of the von-Frey hair to which an animal reacts in 50% of the applications) was calculated.

## Gene silencing via small hairpin RNA

HEK cells expressing Na$_v$1.7 were prepared as described previously (*Cummins et al., 1998*). Cells were cultured in Dulbeccos's modified eagle medium (DMEM)/F12 (Life Technologies, cat# 10565018; Carlsbad, California, USA) containing 10% fetal calf serum, 1% PenStrep (Life Technologies, cat# 15140–122; Carlsbad, California, USA), and 0.6 mg/ml Geneticin (G418, Life Technologies cat#10131035; Carlsbad, California, USA). To knock down the α-GAL gene expression, cells were transfected with small hairpin RNA from the MISSION shRNA Bacterial TRC2 library (Sigma-Aldrich, Taufkirchen, Germany). TRC2 pLKO.5-puro non-mammalian shRNA (SHC202) was used as a control. TRC2-pLKO-puro vector containing shRNA sequence CCGGGATTCGCCAGCTAGCTAATTAC TCGAGTAATTAGCTAGCTGGCGAATCTTTTTG (Clone ID:NM_000169.2–458 s21c1) was amplified and transfected into HEK cells using lipofectamine 3000 transfection reagent (Life Technologies, cat# L3000008, Carlsbad, California, USA). Cells were transfected according to the manufacturer protocol in a six-well plate. Cells were co-transfected with shRNA plasmid and a plasmid expressing green fluorescent protein. HEK cells were incubated in DMEM/F12 medium containing transfection medium for three days (37°C, 5% CO$_2$). Transfection was repeated and cells were incubated for another three days. Cells transfected with shRNA and those with non-mammalian shRNA as a control were used for patch-clamp analysis and immunocytochemistry. We then treated transfected HEK cells with 1.32 μl (1 mg/ml) agalsidase-α (Shire, Saint Helier, UK) and 250 μM lucerastat (N-butyl-deoxy-galactonojirimycin, Biomol, cat# Cay19520-1, Hamburg, Germany) to investigate, if functional ion channel alteration by Gb3 is reversible. Agalsidase-α is used as biweekly intravenous enzyme replacement therapy to treat patients with FD (*Eng et al., 2001*). Lucerastat is an inhibitor of gluco-sylceramide synthase and provides a new therapeutic approach for Fabry disease patients (*Guérard et al., 2018*; *Welford et al., 2018*). Transfected HEK cells were incubated for 24 hr before patch-clamp analysis.

## Immunocytochemistry

To visualize Gb3 deposits in HEK cells, antibodies against CD77 (i.e. Gb3, rat, 1:250, Bio-Rad, cat#; Hercules, California, USA) were used. We applied Alexa Fluor 488 anti-rat IgM (1:300; Jackson Laboratory; Bar Habor, Maine, USA) as secondary antibody together with 4′,6-diamidino-2-phenylindole (1:10.000; Sigma-Aldrich, cat# 28718-90-3, Taufkirchen, Germany). Photomicrographs were assessed manually (Axiophot two microscope, Zeiss, Oberkochen, Germany) using Spot Advanced Software (Windows Version 5.2, Diagnostic Instruments, Inc, Sterling Heights, USA).

## Statistical analysis

Statistical analysis and graph design were performed using SPSS software Version 23 (IBM, Ehningen, Germany) and GraphPad PRISM Version 5.03 (GraphPad Software, Inc., La Jolla, CA, USA). Data distribution was tested using the Kolmogorov-Smirnov test. The non-parametric Mann-Whitney U test for group comparisons was applied, since data were not normally distributed. Behavioral data were analyzed using a two-way ANOVA followed by Tukey's post-hoc test after data transformation applying Johnson's procedure. Data are expressed as line charts representing the mean and standard error of the mean. All other data are visualized as box plots representing the median value and the upper and lower 25% and 75% quartile and bar graphs representing the mean and standard error of the mean as appropriate. p-values<0.05 were considered significant.

## Acknowledgements

We thank Lydia Biko, Helga Brünner, Katharina Gerber, Franziska Karl, Katharina Meder, Sonja Mildner, and Daniela Urlaub for technical assistance. The study was financially supported by research funds of the Interdisciplinary Center for Clinical Research (Interdisziplinäres Zentrum für Klinische Forschung, IZKF) of the University of Würzburg, Germany (NÜ, EW: N-260). NÜ was supported by the German Research Foundation (Deutsche Forschungsgemeinschaft, DFG: UE 171-5/1)

## Additional information

### Funding

| Funder | Grant reference number | Author |
|---|---|---|
| Interdisziplinäres Zentrum für Klinische Forschung, Universitätsklinikum Würzburg | N-260 | Erhard Wischmeyer Nurcan Üçeyler |
| Deutsche Forschungsgemeinschaft | UE 171-5/1 | Nurcan Üçeyler |

The funders had no role in study design, data collection and interpretation, or the decision to submit the work for publication.

### Author contributions

Lukas Hofmann, Formal analysis, Investigation, Methodology, Writing—original draft; Dorothea Hose, Anne Grießhammer, Robert Blum, Formal analysis, Investigation, Writing—review and editing; Frank Döring, Investigation, Writing—review and editing; Sulayman Dib-Hajj, Stephen Waxman, Methodology, Writing—review and editing; Claudia Sommer, Conceptualization, Data curation, Investigation, Writing—original draft; Erhard Wischmeyer, Data curation, Formal analysis, Funding acquisition, Investigation, Methodology, Writing—original draft; Nurcan Üçeyler, Conceptualization, Data curation, Formal analysis, Supervision, Funding acquisition, Investigation, Methodology, Writing—original draft, Project administration

### Author ORCIDs

Lukas Hofmann (iD) http://orcid.org/0000-0002-8397-1819
Sulayman Dib-Hajj (iD) http://orcid.org/0000-0002-4137-1655
Nurcan Üçeyler (iD) http://orcid.org/0000-0001-6973-6428

### Ethics

Animal experimentation: Our study was approved by the Bavarian State authorities (Regierung von Unterfranken, # 54/12).

### Decision letter and Author response

Decision letter https://doi.org/10.7554/eLife.39300.013
Author response https://doi.org/10.7554/eLife.39300.014

## Additional files

### Supplementary files

• Transparent reporting form
DOI: https://doi.org/10.7554/eLife.39300.011

All data generated or analysed during this study are included in the manuscript and supporting files.

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
