## [Decision Letter]

Thank you for submitting your article “Neuronal globotriaosylceramide deposition underlies small fiber pathology in Fabry disease” for consideration by eLife. Your article has been reviewed by three peer reviewers, and the evaluation has been overseen by a Reviewing Editor and Jonathan Cooper as the Senior Editor. The following individuals involved in review of your submission have agreed to reveal their identity: Raphael Schiffmann (Reviewer #1); Anne-Louise Oaklander (Reviewer #2).

The reviewers have discussed the reviews with one another and the Reviewing Editor has drafted this decision to help you prepare a revised submission. We are very interested in your findings and look forward to receiving the revision.

Also, in the course of your revision, please consider a suggestion that been made to revise the title, which is somewhat trivial. Gb3 deposition is certainly the cause of the neuropathy. That is the basic defect in the disease. Perhaps consider 'characterization of the small-fiber neuropathy in a model of Fabry disease'.

*Reviewer #1*

1) The main concern about this paper is the dissimilar genotype between the Fabry mouse and the wild type controls. The Fabry mouse they used are those obtained from NIH and we know they are on a mixed background typically 75% C57BL6 and 25% SVJ129 while the WT are "pure" C57BL6 they generated from female KO something that is very unclear. We are not told that they backcrossed the KO mouse for 10-11 generations into C57BL6 mice to obtain a pure strain. One cannot over-emphasize the critical importance of having identical genetic background in WT and controls. We know that there is a difference in the severity of "disease expression" between the two. Therefore, the authors have to prove to us that this is identical genetic background and if it is not, the results in this paper are not valid. There are numerous examples of papers with false conclusions including in the Fabry field Rozenfeld et al., 2011 that wrongly concluded that cardiac mass in Fabry mouse is significantly lower than wild type mouse. For an overview of this problem see: https://www.jax.org/news-and-insights/2006/june/the-importance-of-genetic-background-in-mouse-based-biomedical-research

2) A second major and basic methodological problem is that we are not told what sex of mouse they use in any of the experiments/figures. They seem to lump together males (presumed hemizygous) and females of unknown genotype – heterozygous or homozygous. The difference would be absolute huge between the heterozygous or homozygous females. The former are not likely to show any abnormality. In general they don't pay attention to the sex of the mouse either WT or KO, which is a big problem.

3) It is important to demonstrate that males and homozygote females are indeed the same. They need to clearly emphasize that these are homozygote females.

4) Use the term 'sex' not 'gender'

5) Figure 6 experiment is not convincing. shRNA Gb3 staining looks not materially different from shRNA. So it is not demonstrated that Gb3 elevation is responsible for the decrease in current. There should be a more direct demonstration that reduction in Gb3, for example using GZ-161 (inhibitor of glucosylceramide synthase) reverses these current abnormalities.

*Reviewer #2:*

Fabry is a rare serious X-linked lysosomal storage disorder in which loss or reduction of activity of by α-22 galactosidase A deficiency damages the central and peripheral nervous systems, heart, and kidneys associated with accumulation of globotriaosylceramide in lysosomes. Genetic testing, animal models, and enzyme replacement therapy are available.

The small-fiber peripheral neurons are among the first cells affected in Fabry, particularly the Aδ thinly myelinated nociceptive fibers, and the first symptoms of Fabry, even in childhood, with widespread neuropathic pain and hypohydrosis. Pain can flare when patients become hot.

It has been previously established that α-GAL knock-out mice (GLA-KO) accumulate globotriaosylceramide (Gb3) in dorsal root ganglion (DRG) neurons. Here the authors compared young and old WT and KO mice to test whether and how Gb3 accumulation might cause neuropathic pain symptoms and signs by studying 3 of the major ion channels that contribute to action potentials in nociceptive neurons, the TRPV1, HCN2, and Na_v_1.7 channels.

This manuscript documents elegant and exhaustive pathological, electrophysiological and behavioral studies that were well controlled and with appropriate data analysis and interpretation. They showed pathologically that in KO mice, Gb3 accumulates within and around the cell bodies of sensory neurons in the dorsal root ganglia in an age-dependent manner and that distal axons of these neurons become progressively depleted in the skin they innervate. The function of TRPV1 did not appear affected, as recordings from cultured DRG neurons from old and young KO and WT mice did not show changes in TRPV1 currents. In contrast, studying the HCN2 pacemaker ion channel, showed behavioral and patch-clamp evidence of loss of HCN2 function in older KO mice and investigating Na_v_1.7 showed reduced Na current densities only in older KO mice with associated changes in pain behaviors, including in standard experimental models of neuropathic pain. Lastly the author used RNA interference to silence α-GAL in HEK cells expressing Na_v_1.7 and showed that this reduced sodium currents, and that enzyme replacement alleviated this decrease.

Although these studies don't clarify the exact mechanisms of how Gb3 affects Na_v_1.7 channels, nor do they clarify if other not-studied ion channels are affected, they highlight the importance of Gb3 and how it damages sensory neurons and their functions. In addition to learning more about these channels, their work raises the question of whether Fabry symptoms and damage might also be improved by pharmacological efforts to reduce Gb3. Enzyme replacement therapy is not universally available, nor continuously active with periodic dosing. Expression of Na_v_1.7 is largely restricted to sensory and sympathetic neurons, but studying other ion channels, particularly other more widely expressed Na_V_ isoforms should be conducted in the other Fabry-affected organs. Both the kidney and heart rely extensively on ion channels as well.

*Reviewer #3:*

The manuscript contains novel and interesting data. At present, the data does not unequivocally support the conclusions and additional data are required.

Methods

The authors should provide details of DRG culture medium composition, particularly glucose content, given that excess glucose can modify TRPV1 expression and function in vitro and in vivo. It is important to appreciate whether these studies were performed at glucose levels consistent with prior in vivo environment or during a period of acute hyperglycemic exposure.

Data

Figure 1: It is important to know location of Gb3 in cross sections of nerve trunk and in skin, not just DRG, particularly given reports of extracellular Gb3 in skin of patients with Fabry disease. The present focus on the DRG alone distorts interpretation of the data. Tissue appears to be available (see Figure 2).

Figure 3: The authors should determine whether the increase in TRPV1 +ve cell bodies occur across all sizes – image 3E appears to illustrate novel expression in relatively large cell bodies. This is important to the authors attempts to directly associate DRG TRPV1 expression with specific pain modalities and can be accomplished on the same tissue sections used to generate the images presented in Figure 3B-E.

Figure 3G-H: The authors attempts to causally link accumulation of Gb3 in sensory neuron cell bodies with apoptosis and loss of axonal projections can be addressed in these DRG neuron cultures by reporting total cell survival rates, apoptotic markers and neurite outgrowth between cultures from WT and GLA-KO mice.

Figure 4J: This graph is interpreted in text as showing post-CCI hyperalgesia in WT mice that is absent in GLA-KO mice, but could equally be interpreted as showing post-CCI loss of sensation in GLA-KO mice vs baseline. The statistical analysis is by Mann-Whitney U test, which only provides an unpaired comparison between the groups at each time point. I am not sure this is valid per se and it also prevents comparison vs baseline within each group. It would be more informative to see a repeat measures analysis vs baseline across time within each group as this would describe whether hyper or hypoalgesia occurred.

Figure 5G: The curve of old GLA-KO mice looks displaced from the WT curve rather than different, because the baseline was higher for the GLA-KO mice. Calculated as% change from baseline within each group, there may be little difference between groups. More caution may be needed in interpretation of this data set.

---

## [Author Response]

Also, in the course of your revision, please consider a suggestion that been made to revise the title, which is somewhat trivial. Gb3 deposition is certainly the cause of the neuropathy. That is the basic defect in the disease. Perhaps consider 'characterization of the small-fiber neuropathy in a model of Fabry disease'.

We have followed the editor’s suggestion and have changed the title as follows:

“Characterization of small fiber pathology in a mouse model of Fabry disease”.

Reviewer #11) The main concern about this paper is the dissimilar genotype between the Fabry mouse and the wild type controls. The Fabry mouse they used are those obtained from NIH and we know they are on a mixed background typically 75% C57BL6 and 25% SVJ129 while the WT are "pure" C57BL6 they generated from female KO something that is very unclear. We are not told that they backcrossed the KO mouse for 10-11 generations into C57BL6 mice to obtain a pure strain. One cannot over-emphasize the critical importance of having identical genetic background in WT and controls. We know that there is a difference in the severity of "disease expression" between the two. Therefore, the authors have to prove to us that this is identical genetic background and if it is not, the results in this paper are not valid. There are numerous examples of papers with false conclusions including in the Fabry field Rozenfeld et al., 2011 that wrongly concluded that cardiac mass in Fabry mouse is significantly lower than wild type mouse. For an overview of this problem see: https://www.jax.org/news-and-insights/2006/june/the-importance-of-genetic-background-in-mouse-based-biomedical-research

We apologize for not being clear in this point. We paid particular attention and worked very carefully on generating littermate wildtype (WT) mice for our experiments and confirm that we exclusively used mice of identical genetic background. We did not perform our experiments comparing α-galactosidase A knockout (GLA KO) mice with “"pure" C57BL6” mice as assumed by the reviewer. We started our mouse colony with GLA KO mice, as stated and referenced (Ohshima, 1997) in our manuscript. We are aware of the mixed background with C57BL6 and SVJ129 and did not intend to generate GLA KO mice with pure C57BL6 background by multiple back-crossing. To ensure that our KO and WT mice have an identical genetic background, we crossed these GLA KO mice with C57BL6/N mice to first generate heterozygous offspring. These heterozygous mice were then cross-bred with each other to obtain homozygous female and male GLA KO and WT mice. In the further course of breeding, we mated these two homozygous lines only with genetically matching mice (KO x KO, WT x WT) of the respective strain. Thus, we generated two mouse strains (homozygous GLA KO and WT) with an identical genetic background. To maintain the purity of these strains, we performed genotype analysis on every single mouse born in our animal facility. We have included this detailed information in the Materials and methods section of our revised manuscript:

“We used 95 GLA KO mice (45 male, 50 female) of mixed genetic background (C57BL6 and SVJ129) carrying a targeted disruption of the α-GAL gene as previously described (Ohshima et al., 1997). Additionally, 96 WT littermate mice (45 male, 51 female) were assessed. To ensure that our KO and WT mice have an identical genetic background, we first crossed GLA KO mice with C57BL6/N mice to generate heterozygous off-springs. These heterozygous mice were then cross-bred with each other to obtain homozygous female and male GLA KO and WT mice. In the further course of breeding, we mated these two homozygous lines only with genetically matching mice (KO x KO, WT x WT) of the respective strain. Thus, we generated two mouse strains (homozygous GLA KO and WT) with an identical genetic background. To maintain the purity of these strains, we performed genotype analysis on every single mouse born in our animal facility. For genotyping, we used the Kapa2G fast PCR Kit (Kapa Biosystems, Wilmington, USA) and the following primers: oIMR5947, AGGTCCACAGCAAAGGATTG; oIMR5948, GCAAGTTGCCCTCTGACTTC; oIMR7415, GCCAGAGGCCACTTGTGTAG.”

2) A second major and basic methodological problem is that we are not told what sex of mouse they use in any of the experiments/figures. They seem to lump together males (presumed hemizygous) and females of unknown genotype – heterozygous or homozygous. The difference would be absolute huge between the heterozygous or homozygous females. The former are not likely to show any abnormality. In general they don't pay attention to the sex of the mouse either WT or KO, which is a big problem.

Sex differences are indeed crucial and we have paid particular attention to this issue in our GLA KO mouse colony. For this reason, we first performed a separate extensive three-year study assessing young and old (3 to >24 months) male and female GLA KO (89 male, 126 female) and WT littermate mice (80 male, 46 female) of identical genetic background for their age and gender specific development using a broad array of behavioral tests (Üçeyler et al., 2016). We showed that while pain behavior changed with age, there was no difference between male and female mice at any age. This result is not surprising though, since the GLA KO mice used were generated by targeted disruption of the α-galactosidase A (α-GAL) gene, thus both male and female GLA KO mice are complete knockouts with no α-GAL enzyme activity: male mice as hemizygotes and female mice always as homozygotes. The human situation with heterozygote female individuals is thus different to the female GLA KO mouse. This information was already given in our revised manuscript, please see subsection “Mice and study groups”:

“Animals were not stratified for gender, since in GLA KO mice α-GAL knockout results in a complete loss of enzyme activity in both sexes (i.e. homozygous female mice, hemizygous male mice) with similar pain behavior in contrast to human patients (Lakoma et al., 2014; Lakoma et al., 2016; Rodrigues et al., 2009; Üçeyler et al., 2016).”

Based on these extensive data (Üçeyler et al. 2016), we here studied both male and female GLA KO and WT mice. The numbers of male and female mice used were already given in the Methods section of our manuscript, please see subsection “Mice and study groups”:

“We used 95 GLA KO mice (45 male, 50 female) of mixed genetic background (C57BL6 and SVJ129) carrying a targeted disruption of the α-GAL gene as previously described (Ohshima et al., 1997). Additionally, 96 WT mice (45 male, 51 female) were assessed.”

The respective information was also given in the figure legends of the behavioral

experiments. We have now also included the sex distribution of all other experimental mice in the respective Figure legends in the revised version of our manuscript

3) It is important to demonstrate that males and homozygote females are indeed the same. They need to clearly emphasize that these are homozygote females.

Please see our reply to question one above. We have included this information in the Materials and methods section of our revised manuscript.

4) Use the term 'sex' not 'gender'

We have followed the reviewer`s suggestion and have exchanged the wording in subsection “Mice and study groups” of our revised manuscript.

5) Figure 6 experiment is not convincing. shRNA Gb3 staining looks not materially different from shRNA. So it is not demonstrated that Gb3 elevation is responsible for the decrease in current. There should be a more direct demonstration that reduction in Gb3, for example using GZ-161 (inhibitor of glucosylceramide synthase) reverses these current abnormalities.

We have specifically inhibited the α-galactosidase A (α-GAL) using shRNA. While Na_v_1.7 currents were normal at baseline, enzyme inhibition led to an increase in cellular Gb3 which we demonstrate immunocytochemically. This increase was paralleled by a drop in Na_v_1.7 currents. Using agalsidase-α as a specific compound, cellular Gb3 load was decreased and Na_v_1.7 currents were restored. We believe that this experiment does demonstrate the causal link between cellular Gb3 load and Na_v_1.7 currents, even though not clarifying the exact mechanism of interaction. We have repeated our experiment and have updated former Figure 6 (now Figure 7) accordingly. We also followed the Reviewer`s suggestion and have performed an additional experiment in our human embryonic kidney 293 (HEK) cell system using an inhibitor of glucosylceramide synthase (N-butyldeoxygalactonojirimycin, i.e. lucerastat, Biomol, cat# Cay19520-1, Hamburg, Germany). In analogy to our results after incubation with agalsidase-α (Figure 7G-I, N), incubation of shRNA treated cells with lucerastat for 24 hours resulted in a marked reduction of intracellular Gb3 paralleled by recovery of Na_v_1.7 currents. We have included this information in the Materials and methods section, Results section and Discussion section of our revised manuscript data are illustrated in the updated Figure 7 (former Figure 6):

“Gb3 accumulation and reversible reduction of Na_v_1.7 currents in HEK cells after shRNA treatment

Finally, we investigated if cellular Gb3 accumulation interferes with Na_v_1.7 currents. […] Electrophysiological analysis of Na_v_1.7 currents in Gb3-positive HEK cells revealed a marked decrease of sodium currents after shRNA treatment compared to control HEK cells (p<0.01, Figure 7J, K), which recovered after agalsidase-α and lucerastat incubation (agalsidase-α: p<0.05; lucerastat: p<0.01, Figure 7N).”

Discussion section:

“Most intriguingly, this reduction of Na_v_1.7 currents was reproducible in vitro by silencing α-GAL in HEK cells and was reversed by treatment of HEK cells with the Fabry-specific drug used as enzyme replacement therapy in Fabry patients, agalsidase-α and a new Gb3 synthase inhibitor lucerastat (Figure 7), pointing to a functional effect of Gb3 on Na_v_1.7 channels.”

Materials and methods section:

“We then treated transfected HEK cells with 1.32 µl (1mg/ml) agalsidase-α (Shire, Saint Helier, UK) and 250 µM lucerastat (N-butyldeoxygalactonojirimycin, Biomol, cat# Cay19520-1, Hamburg, Germany) to investigate, if functional ion channel alteration by Gb3 is reversible. Agalsidase-α is used as biweekly intravenous enzyme replacement therapy to treat patients with FD (Eng et al., 2001). Lucerastat is an inhibitor of glucosylceramide synthase and provides a new therapeutic approach for Fabry disease patients (Guerard et al., 2018; Welford et al., 2018). Transfected HEK cells were incubated for 24 hours before patch-clamp analysis.”

Reviewer #2:

*[…] Although these studies don't clarify the exact mechanisms of how Gb3 affects* Na_v_1.7 *channels, nor do they clarify if other not-studied ion channels are affected, they highlight the importance of Gb3 and how it damages sensory neurons and their functions. In addition to learning more about these channels, their work raises the question of whether Fabry symptoms and damage might also be improved by pharmacological efforts to reduce Gb3. Enzyme replacement therapy is not universally available, nor continuously active with periodic dosing. Expression of* Na_v_1.7 *is largely restricted to sensory and sympathetic neurons, but studying other ion channels, particularly other more widely expressed Na_V_ isoforms should be conducted in the other Fabry-affected organs. Both the kidney and heart rely extensively on ion channels as well.*

We thank the reviewer for this appreciative comment on our work and fully agree that our study still leaves essential questions unanswered. We have further accomplished the respective passage stating on these issues in the Discussion section of the revised manuscript:

“The exact mechanisms, however, remain to be elucidated, we show that neuronal Gb3 deposits result in […]”

“Our study also underscores the importance of investigating further neuronal ion channels like Na_v_ and HCN isotypes and of studies in other organ systems, such as the heart and kidneys, to better understand the effect of Gb3 on e.g. cardiomyocytes in the generation of lethal arrhythmias.”

Reviewer #3:The manuscript contains novel and interesting data. At present, the data does not unequivocally support the conclusions and additional data are required.Methods

*The authors should provide details of DRG culture medium composition, particularly glucose content, given that excess glucose can modify TRPV1 expression and function* in vitro *and* in vivo*. It is important to appreciate whether these studies were performed at glucose levels consistent with prior* in vivo *environment or during a period of acute hyperglycemic exposure.*

We used TNB-100 medium (Biochrom, cat# F8023; Berlin, Germany), containing 25 mM glucose, for cultivation of DRG neurons. Murine blood glucose levels are about 8 mM (Klueh et al., 2006). TNB-100 medium is widely used as a standard medium for cultivation of primary neuronal cells (for references see e.g. (de la Roche et al., 2016; Eberhardt et al., 2017a; Obreja et al., 2005; Vetter et al., 2012)) and is not different compared to e.g. Neurobasal Medium (Thermo Fisher, cat# 21103049, Waltham, Massachusetts, USA). It is indeed known that hyperglycemia can lead to increased TRPV1 currents (Lam et al., 2018), however, we did not detect any inter-genotype differences (Figure 4K). Furthermore, DRG neurons of both genotypes and age-groups were cultured with the same amount of TNB-100 medium without adding further glucose. For patch clamp analysis, a bath solution with 10 mM glucose was used, which is suitable to measure TRPV1 currents (Eberhardt et al., 2017b). Following the reviewer`s suggestion we have extended the information on DRG culture medium and bath solution composition in the Materials and methods section of the revised manuscript:

“Mouse DRG neurons were dissected and cultivated in culture medium (100 ml TNB-100, Biochrom, cat# F8023; Berlin, Germany, 25 mM glucose; 2 ml PenStrep, Life Technologies, cat# 15140-122; Carlsbad, CA, USA; 100 µl L-glutamine, Life Technologies, cat# 25030-032; Carlsbad, CA, USA; 2 ml protein-lipid-complex, Biochrom, cat# F8820; Berlin, Germany) containing 25 ng/ml nerve growth factor (2.5S, Alomone Labs, cat# N-240; Jerusalem, Israel) according to a previously published protocol (Langeslag et al., 2014).”

"135 mM NaCl, 5.4 mM KCl, 1.8 mM CaCl2, 1 mM MgCl2, 10 mM glucose, and 5 mM HEPES (Eberhardt et al., 2017b; Hamill et al., 1981).”

DataFigure 1: It is important to know location of Gb3 in cross sections of nerve trunk and in skin, not just DRG, particularly given reports of extracellular Gb3 in skin of patients with Fabry disease. The present focus on the DRG alone distorts interpretation of the data. Tissue appears to be available (see Figure 2).

We followed the reviewer`s suggestion and have performed additional experiments immunoreacting mouse sciatic nerve and skin sections with antibodies against CD77 for clarifying the extent of Gb3 deposition outside the DRG. As in WT mice, we did not find Gb3 deposits in sciatic nerve preparations of old GLA KO mice. Hence, extraneuronal Gb3 deposits seem to be restricted to the very proximal parts of the axons still within the DRG, as we demonstrated in our supplementary Video in the original version of our manuscript. Gb3 was also absent in skin samples of GLA KO and WT mice. We have included these additional experiments in the Materials and methods section and Results section of our revised manuscript and also provide the respective Results in Figure 2F-Q and its legend:

Results section:

“Furthermore, we assessed whether Gb3 also accumulates in axons of the sciatic nerve and in skin. We did not find any Gb3 depositions in the sciatic nerve (Figure 2F-K) or in footpad skin (Figure 2L-Q) of old WT and GLA KO mice.”

Materials and methods section:

“Ten-µm DRG and sciatic nerve cryosections were prepared with a cryostat (Leica, Bensheim, Germany).”

“Additionally, antibodies against β-(ΙΙΙ)-tubulin (chicken, 1:500, Abcam, cat# ab41489, Cambridge, UK), BiP (rabbit, 1:5000, Abcam, cat# ab21685, Cambridge, UK) and CD77 (i.e. Gb3, rat, 1:250, Bio-Rad, cat# MCA579; Hercules, California, USA) were used to document endoplasmic stress responses under pathophysiological conditions (Lee, 2005). We used goat anti-rabbit IgG, and rabbit anti-goat IgG and goat anti-chicken IgG labelled with cyanine 3.18 fluorescent probe (1:50, Amersham; Piscataway, New Jersey, USA) and Alexa Fluor 488 anti-rat IgM (1:300; Jackson Laboratory; Bar Habor, Maine, USA) as secondary antibodies.”

Figure 3: The authors should determine whether the increase in TRPV1 +ve cell bodies occur across all sizes – image 3E appears to illustrate novel expression in relatively large cell bodies. This is important to the authors attempts to directly associate DRG TRPV1 expression with specific pain modalities and can be accomplished on the same tissue sections used to generate the images presented in Figure 3B-E.

We have re-analyzed our DRG preparations immunoreacted with antibodies against TRPV1 and CD77 to determine the neuronal cell populations that are mainly TRPV1 positive. We found that TRPV1 immunoreaction was positive mainly in small diameter DRG neurons (i.e. <25 µm in diameter) without inter-group difference and independent of age. We have included this information in the Materials and methods section and Results section of the revised manuscript and also in the revised version of former Figure 3 (now Figure 4) and its legend:

“We also analyzed the distribution of TRPV1 immunoreactivity across different neuronal sizes and quantified TRPV1 positive neuron diameters; neuron populations were stratified as small (<25 µm in diameter) and large (≥25 µm in diameter) neurons (Figure 4G) (Cesare and McNaughton, 1996; Hoheisel et al., 1994; Lawson et al., 1993). TRPV1 immunoreactivity was mainly observed in small diameter neurons independent of genotype and age.”

Gb3 deposition leads to an increase in neuronal size, which we had already shown for cultivated neurons of old GLA KO mice in former Figure 4 (now Figure 5) of the original manuscript. For data completion we now also quantified neuronal size in DRG preparations of our mouse cohort. We confirm an increase in cell size as measured by cellular area in young GLA KO compared to young WT mice and in old GLA KO compared to young GLA KO and old WT mice. We have also included these data in the Materials and methods section and Results section of the revised manuscript and in Figure 1 and its legend:

Results section:

“First, we examined DRG neuron size by analyzing neuronal area (Figure 1A-D) and found larger DRG neurons in young GLA KO compared to young WT mice (p<0.01; Figure 1E). Neurons of old GLA KO mice were larger compared to old WT (p<0.001) and young GLA KO mice (p<0.001; Figure 1E).”

Materials and methods section:

“We performed hematoxylin-eosin staining. Briefly, DRG cryosections were incubated in hematoxylin (Sigma-Aldrich, cat# H3136, Taufkirchen, Germany) for 10 min and 25 s with 1% eosin (Sigma-Aldrich, cat# 23251, Taufkirchen, Germany). Afterwards, cryosections were dehydrated with an ascending ethanol row. To quantify cell size, neurons were surrounded using Fiji software (ImageJ 1.50g, Wayne Rasband, National Institute of Health, USA) (Schindelin et al., 2012) and perimeter was calculated.”

Figure 3G-H: The authors attempts to causally link accumulation of Gb3 in sensory neuron cell bodies with apoptosis and loss of axonal projections can be addressed in these DRG neuron cultures by reporting total cell survival rates, apoptotic markers and neurite outgrowth between cultures from WT and GLA-KO mice.

We followed the reviewer`s suggestion and have performed a NucView 488 Caspase 3 Enzyme Substrate Assay to assess apoptosis in cultured DRG neurons. We have additionally determined neurite outgrowth. Cultured DRG neurons of old GLA KO mice showed an increased caspase 3 activity already 48 hours after cultivation compared to old WT neurons. DRG neurons of old GLA KO mice incubated with 500 nM staurosporine as a positive control displayed a higher percentage of caspase 3 positive neurons compared to neurons in the naïve state and WT positive control neurons. Additionally, neurite outgrowth was lower in the GLA KO neurons than in neurons cultured from old WT mice. We have included this information in the Materials and methods section, Results section and Discussion section of the revised manuscript and have also created the new Figure 3 with its legend:

“Increased apoptosis and decreased neurite outgrowth in cultured DRG neurons of old GLA KO mice

To investigate the degree of apoptosis in DRG neurons in the course of Gb3 accumulation and potential endoplasmic stress, we performed a NucView 488 Caspase 3 Enzyme Substrate Assay. We quantified the percentage of caspase 3 positive neurons in cultured DRG neurons of old GLA KO and WT mice (Figure 3 A-D). DRG neuron cultures of old GLA KO mice in the naïve state displayed a higher percentage of caspase 3 positive neurons compared to old WT mice (p<0.001, Figure 3E) indicating enhanced apoptosis. Additionally, positive control neurons of both genotypes incubated with 500 nM staurosporine for 16 hours showed a higher percentage of caspase 3 positive neurons compared to cultured DRG neurons in the naïve state (p<0.05 each, Figure 3 E). We further determined the percentage of neurons with neurite outgrowth. Cultured DRG neurons of old GLA KO mice showed less neurite outgrowth compared to neurons of WT mice (p<0.001, Figure 3F).”

Discussion section:

“Indeed, DRG neurons of old GLA KO mice also displayed increased caspase 3 activity and decreased neurite outgrowth as markers of apoptosis. Increased caspase 3 activity is associated with cellular vulnerability and apoptotic cell death (Hartmann et al., 2000) and is involved in DNA breakdown and morphological changes during apoptosis (Janicke et al., 1998).”

Materials and methods section:

“Caspase 3 substrate assay

DRG neurons of old GLA KO and WT mice, were dissected and cultured for 48 h as described above. To analyze apoptosis, we performed a NucView 488 Caspase 3 Enzyme Substrate Assay (Biotium, cat# 10403, Fenton, California, USA) according to the manufacturer`s protocol. As a positive control, cells of both genotypes were incubated with 500 nM staurosporine (Abcam, cat# ab120056, Cambridge, UK) for 16 h prior to performing the NucView 488 Caspase-3 Enzyme Substrate Assay. For quantification of apoptosis, the percentage of caspase 3 positive neurons and the percentage of neurons with neurite outgrowth was determined.”

Figure 4J: This graph is interpreted in text as showing post-CCI hyperalgesia in WT mice that is absent in GLA-KO mice, but could equally be interpreted as showing post-CCI loss of sensation in GLA-KO mice vs baseline. The statistical analysis is by Mann-Whitney U test, which only provides an unpaired comparison between the groups at each time point. I am not sure this is valid per se and it also prevents comparison vs baseline within each group. It would be more informative to see a repeat measures analysis vs baseline across time within each group as this would describe whether hyper or hypoalgesia occurred.

Withdrawal thresholds of old GLA KO mice at baseline and 3 d after CCI do not differ significantly. The visual impression of higher thresholds in former Figure 4 (now Figure 5) at 3 days after surgery is caused by single outliers in the experimental group, which is also reflected by the higher standard deviation. We followed the reviewer`s suggestion and re-analyzed behavioral data using two-way ANOVA following Tukey’s *post-hoc* test after data transformation using Johnson’s procedure. Results of mechanical sensitivity following CCI did not change except for no intergroup difference comparing old GLA KO and WT mice on day 7 after surgery (originally: p<0.01, now: n.s.). We have updated the respective information in the Materials and methods section of our revised manuscript and in the new Figure 5J:

“Behavioral data were analyzed using a two-way ANOVA followed by Tukey’s post-hoc test after data transformation applying Johnson`s procedure. Data are expressed as line charts representing the mean and standard error of the mean.”

Figure 5G: The curve of old GLA-KO mice looks displaced from the WT curve rather than different, because the baseline was higher for the GLA-KO mice. Calculated as% change from baseline within each group, there may be little difference between groups. More caution may be needed in interpretation of this data set.

We followed the reviewer`s suggestion and calculated von Frey values as percent change from baseline, which shows that there is a slight difference comparing old WT and GLA KO mice. Old GLA KO mice, however, are still less affected than old WT mice (please see Author response image 1). We have modified the respective passage in the Results section of our revised manuscript:

**Author response image 1. respfig1:** Mechanical sensitivity after injection of complete Freund`s adjuvant. Line chart displays mechanical withdrawal thresholds from old (≥12 months) wildtype (WT) and α-galactosidase A deficient (GLA KO) mice, calculated as percent change from baseline. Old WT mice were slightly more sensitive to mechanical stimulus, which was not significant except for 48 hours after CFA injection compared to old GLA KO mice. Data were analyzed using two-way ANOVA following Tukey’s post-hoc test after data transformation using Johnson’s procedure.

Modified Results text:

“Similarly, all mice developed mechanical hypersensitivity starting one hour after CFA injection compared to baseline (p<0.001, Figure 6G), which was less pronounced but not different in old GLA KO mice compared to old WT mice after CFA injection (Fig. 25 6G), and all mice remained mechanically hypersensitive until day seven after CFA injection.”